

# 1  Mechanism and effects of warming water in ice-covered Ngoring
# 2  Lake of Qinghai-Tibet Plateau

Mengxiao Wang [1,2], Lijuan Wen [1*], Zhaoguo Li [1], Matti Leppäranta [3],
Victor Stepanenko [4,5], Yixin Zhao [1,2], Ruijia Niu [1,2], Liuyiyi Yang [1,2] and
Georgiy Kirillin [6]
[1] Key Laboratory of Land Surface Process and Climate Change in Cold and Arid
Regions, Northwest Institute of Eco-Environment and Resources, Chinese Academy of
Sciences, 730000 Lanzhou, China
[2] University of Chinese Academy of Sciences, 10049 Beijing, China
[3] Institute of Atmospheric and Earth Sciences, University of Helsinki
[4] Research Computing Center, Lomonosov Moscow State University, Moscow, Russia
[5] Moscow Center for Fundamental and Applied Mathematics, Moscow, Russia
[6] Department of Ecohydrology, Leibniz-Institute of Freshwater Ecology and Inland
Fisheries (IGB), Berlin, Germany
*Correspondence to: Lijuan Wen (wlj@lzb.ac.cn)
**Abstract.** Ngoring Lake is the largest freshwater lake in the Qinghai-Tibet Plateau (TP).
The lake water temperature was observed to be generally rising during the ice-covered
period from November 2015 to April 2016. This phenomenon appeared in the whole
water column, with slowing in deep water and accelerating in shallow water before ice
melting. The process is different from low-altitude boreal lakes. There are few studies
on its mechanism and effects on lake-atmosphere interaction. Based on the observation
data of Ngoring Lake Station, ERA5-Land data, MODIS surface temperature data, and
precipitation data of Maduo Station of China Meteorological Administration, the
characteristics of water temperature rise in the ice-covered Ngoring Lake are analyzed.
LAKE2.3 model, which is currently little used for TP lakes, is applied to explore the
influence of local climate characteristics and the main physical parameters on the
radiation transfer in water body. The study questions are the continuous rise of water
temperature in the ice-covered period, and the effects of different water temperature
profiles prior to ice breakup on the lake heat storage per unit area and sensible and latent
heat release. The results show that LAKE2.3 represents well the temperature evolution
and thermal stratification in Ngoring Lake, especially in the ice-covered period. The
strong downward short-wave radiation plays a dominant role, low precipitation gives



positive feedback, and smaller downward long-wave radiation, lower temperature and
larger wind speed give negative feedback. Increase of ice albedo and ice extinction
coefficient reduces the heating rate of water temperature before reaching the maximum
density temperature, and increases the maximum temperature that can be reached
during ice-covered period, while increasing the water extinction coefficient has little
influence on water temperature. The lake temperature in Ngoring Lake rising during
the ice-covered period, and the temperature at the upper layer of lake body was higher
than that at the maximum density temperature before ice breaking. Compared with the
characteristics of three typical ice-covered periods which the lake temperature remained
fixed in each layer, and the lake temperature was less than or equal to the maximum
density temperature, the difference of heat release after ice breaking lasted for 59-97
days. The higher the lake temperature before breakup, the more heat is stored in the
lake, and the more sensible heat and latent heat is released when the ice melts
completely and the faster is the heat release.

## 1 Introduction

The Tibetan Plateau (TP), with an average altitude of 4000-5000 m, is known as the
"roof of the world". It is the highest plateau on Earth. There are many alpine lakes in
TP constituting the largest number, largest area and highest altitude plateau lake group
in China, known as the "Asian water tower" (Immerzeel et al., 2010). There are more
than 1400 lakes with an area of more than 1 km$^2$, and the total area of lakes is more
than $5 \times 10^4$ km$^2$, accounting for 57.2 % of the total lake area in China (Wan et al., 2016;
Zhang et al., 2019).
Lakes are sensitive to climate change and therefore act as indicators of climate change
(Adrian et al., 2009; Qin et al., 2009). Under the background of global warming, the
surface temperature of lakes has been rising (O'Reilly et al., 2015; Schmid et al., 2014;
Sharma et al., 2015; Zhang et al., 2014). The warming rate of the TP is twice the global
average. Among the 52 plateau lakes surveyed, the surface temperature of 60 % of the
lakes shows an upward trend, while the surface temperature of some lakes shows a
downward trend due to the melting of glaciers (Duan and Xiao, 2015; Yang et al., 2014;
Zhu et al., 2020). The surface temperature affects the thermal stratification and the
length of the ice-covered period, not only on the stability and vertical convection of the
lake. Also the material and energy exchange between the lake and the atmosphere
depend on the surface temperature (Efremova et al., 2013; Ramp et al., 2015; Rösner et
al., 2012) that has impact on the local climate (Gerken et al., 2013; Li et al., 2016a; Wen
et al., 2015; Xu and Liu, 2015; Yang and Wen, 2012).
At the same time, lake temperature is an important indicator of lake ecosystem



restricting the biochemical process inside the lake. The temperature not only changes
the content of dissolved oxygen, nitrogen, phosphorus and other nutrients but also
changes the rate of biochemical reactions, thus affecting the water quality and
distribution of aquatic organisms in the lake (Dokulil, 2013; Hardenbicker et al., 2016;
Li et al., 2015a; Weitere et al., 2010). Lake ecology is an important part of the Tibetan
Plateau ecosystem. Therefore, it is of great significance to study the characteristics of
lake temperature for understanding the plateau aquatic ecosystem in the local weather
and climate conditions.
Due to the variations of latitude, altitude and depth, the lake temperature characteristics
are different in lakes of different geographical regions. Tropical and subtropical lakes
possess high temperature throughout the year, and in winter, they do not freeze and are
not affected by ice. For example, Kivu Lake, a tropical lake located in central Africa,
has a winter temperature greater than 20 ℃ (Thiery et al., 2014), and Taihu, a
subtropical lake located in the middle and lower reaches of the Yangtze River in China,
and Mangueira Lake, a subtropical lake located in Rio Grande do Sul, have a winter
temperature greater than 5 ℃ (Tavares et al., 2019; Chen et al., 2021). However, the
boreal lakes located in middle and high latitudes usually freeze in winter, and the
increase of the lake surface albedo due to ice cover affects the radiation transfer into
water body and, consequently, the lake temperature (Erm et al., 2010). Among lakes
located in middle and high latitudes but at a low altitude, the under-ice water of some
lakes is evenly mixed in winter, and the temperature of the entire lake is basically
maintained at a temperature between 0-$T_{\rho,max}$ ($T_{\rho,max}$ is the maximum density, $T_{\rho,max}$ =
3.98 °C for fresh water). Such lakes are, for example, Sunapee Lake, a northern
temperate lake in New Hampshire, Simcoe Lake in southern Ontario, Canada, and
Mendota Lake in the United States (Bruesewitz et al., 2015; Yang et al., 2017; Yang et
al., 2021). Another class of mid- and high-latitude lakes have an inverse winter
stratification. The ice-water interface temperature is at the freezing point, and the deeper
water temperature is the highest, at most $T_{\rho,max}$, and the temperature in each layer is
maintained at a temperature between the freezing point and $T_{\rho,max}$. Temperature
increases from the upper layer to the lower layer in such lakes as Thrush Lake in
Northestren Minnesota (Fang and Stefan, 1996), Pääjärvi Lake located in southern
Finland (Saloranta et al., 2009) and Valkea-Kotinen Lake (Li et al., 2016b). Their
vertical temperature profiles are different, but the temperature is between the freezing
point and $T_{\rho,max}$ during ice-covered period. Therefore, we present the question: what are
the winter temperature characteristics of high-altitude lakes?
Due to the harsh environment of the TP and difficulties in collecting field observations,
field studies have been limited to several large lakes such as Nam Co, Bangong Co,
Qinghai Lake, Ngoring Lake, and most of them have been performed in ice-free period



(Lazhu et al., 2015; Su et al., 2020; Huang et al., 2019; Song et al., 2020). It has been
found out that in several lakes in TP the temperature rises during the ice-covered period.
The temperature of Bangong Co and Nam Co Lakes have risen from freezing to melting,
but the rise is greater in the latter due to the difference in lake depth. The temperature
in Dagze Co Lake remained fixed in each layer in the early ice-covered period and
began to rise in the late ice-covered period, because this lake is meromictic with high
salt content (Wang et al., 2014; Lazhu et al., 2021; Wang et al., 2021). Ngoring Lake is
the largest and relatively shallow freshwater lake on the TP. Its temperature has been
rising throughout the whole ice-covered period, and studies show that solar radiation
transfer plays an important role in this process (Wang et al., 2021; Kirillin et al., 2021).
Although previous studies have revealed the warming phenomenon of TP lakes during
ice-covered period with qualitative analysis pointing out that temperature rise is
affected by salinity and depth (Lazhu et al., 2021), they did not consider the influence
of ice, meteorological conditions, and physical processes in the warming.
Numerical models are often used to reveal the phenomena and mechanisms of TP lakes.
At present, the lake models widely used in the TP are the Flake model (Freshwater Lake
Model) and the lake scheme coupled in the CLM model (Community Land Model)
CoLM (Common Land Model), and WRF (Weather Research and Forecasting Model)
(Fang et al., 2017; Lazhu et al., 2016; Wen et al., 2016; Song et al., 2020; Dai et al.,
2018; Huang et al., 2019; Wu et al., 2021). However, the simulated water temperatures
in winter by the two models kept stationary and could not reproduce the observed rising
temperature (Lazhu et al., 2016; Wen et al., 2016; Huang et al., 2019).
This paper 1) applies the LAKE model for a typical TP lake to evaluate its capability to
simulate the rising lake temperature in ice-cover period; 2) uses the LAKE model to
analyze the influence of the meteorological driving factors and the main parameters that
affect the radiation transmission on the warming process during the ice season in
Ngoring Lake; 3) discusses the influence of temperature distribution prior to ice
breakup on lake heat storage and lake-air heat transfer.

**2 Study Area and Data**
**2.1 Study Area**
Ngoring Lake (34.76º N-35.08º N, 97.53º E-97.90º E, Fig. 1) is located in the western
valley of Maduo County on the eastern TP, with an average lake surface elevation of
4274 m a.s.l. It is the largest freshwater lake in the Yellow River source region. Its
surface covers an area of 610 km$^2$, and the maximum and average depth are 32 and 17
m, respectively. The pH is 8.49 and there are only few fish in the lake. Aquatic plants




grow only in the riparian area. The lake is thermally stratified in summer and ice-
covered from the end of November or early December to late April (Wen et al., 2016).
Ngoring Lake basin is dominated by the cold, semi-arid continental climate, which is
sensitive to the Westerly jet, Indian monsoon and Asian monsoon (Zhang et al., 2013).
According to observation data from 1953 to 2016 at Maduo Station (34.9º N, 98.2º E)
of China Meteorological Administration, the average annual precipitation was 322.4
mm, mostly concentrated in May to September. The average annual air temperature was
-3.53 ºC, and the maximum and minimum air temperature were 24.3 ºC and -48.1 ºC,
occurred in July 20, 2006 and January 2, 1978, respectively.

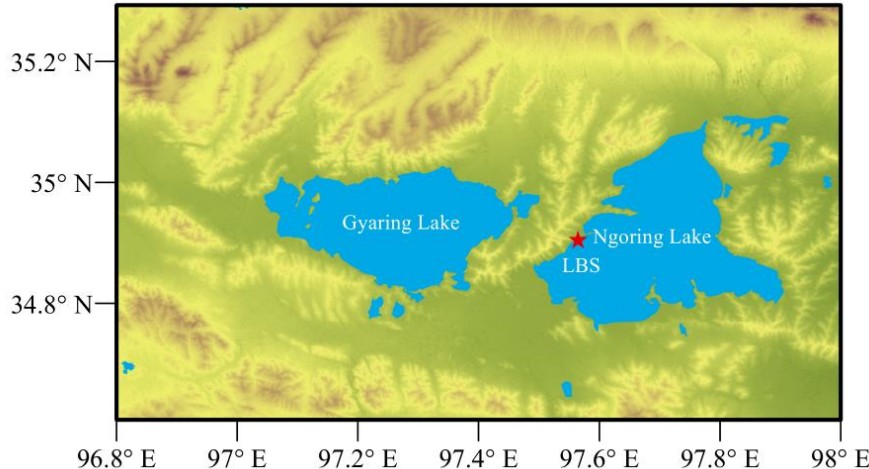

**Figure 1. Location of Ngoring Lake, and the pentagram denotes the lake border**
**station (LBS).**

**2.2 Data**
**2.2.1 LBS Station Data**
The long-term autonomous lake border station (LBS) has an altitude of 4282 m above
sea level, installed at 34.91º N and 97.55º E in October 2012 (Fig. 1). It provided 10 m
altitude wind speed, 2 m altitude air temperature, specific humidity and air pressure,
and 1.5 m altitude downward shortwave (SR) and longwave radiation (LR) from
September 22, 2015 to September 22, 2016 as the driving data for the model (Li et al.,
2020). Li et al. (2015b) and Wen et al. (2016) are referred for the detailed information
about the site configuration and measured quantities. Precipitation intensity was
calculated according to the accumulated precipitation of Maduo Station from 20:00 to
20:00 the next day in the daily value data set (V3.0) of Chinese surface climate data
(http://data.cma.cn). The water temperature profile was observed in the northern part of
Ngoring Lake, where the water depth was 23-25 m in 2015-2016 (Li et al., 2020).

**2.2.2 MODIS Lake surface temperature**
The Land Surface Temperature 8-Day L3 Global product (MYD11C2), which is
derived from the data of Moderate Resolution Imaging Spectroradiometer (MODIS),
was used to determine the ice-covered period in the Ngoring Lake and to verify the
simulated results, due to the lack of ice thickness and water surface temperature
observations. MODIS provides daily global coverage with high spatial resolution on a
long-term basis. The product which is obtained by synthesizing and averaging values
from the corresponding eight MYD11C2 daily files provides the land surface
temperature (LST) at a resolution of 0.05º latitude/longitude (5600 m at the equator)
for Climate Modeling Grid (CMG) (https://ladsweb.nascom.nasa.gov/search) (Wan et
al., 2004).

**2.2.3 ERA5-Land Data**
ERA5-Land is produced as an enhanced global dataset for the land component of the
fifth generation of European ReAnalysis (ERA5) by the European Centre for Medium-
Range Weather Forecasts (ECMWF), framed within the Copernicus Climate Change
Service (C3S) of the European Commission. It is available for hourly ERA5-Land
record for 40 years from 1981 to the present, and ERA5-Land back extension (1950-
1980) is in preparation. Compared to ERA5 (31 km) and ERA-Interim (80 km), ERA5-
Land has enhanced horizontal resolution of 9 km (~0.08º). It is convenient for users to
interpolate data into the longitude and latitude grid of 0.1º spacing
(https://cds.climate.copernicus.eu/cdsapp#!/dataset/reanalysis-era5-land?tab=form)
(Hersbach et al., 2020; Muñoz-Sabater et al., 2021).

**3 Methods**
**3.1 LAKE Model**
The one-dimensional lake model LAKE, including thermodynamic, hydrodynamic and
biogeochemical processes, is used to solve the horizontally averaged transfer equations
of gases, heat, salinity and momentum in an enclosed water body (Stepanenko et al.,
2016; Stepanenko et al., 2011). The vertical heat transfer is simulated, and the
penetration of SR in water layers (Heiskanen et al., 2015), ice, snow and bottom
sediments (Cao et al., 2020) is taken into account. The exchange between water and
inclined bottom is modelled explicitly because the model equations have been averaged
over horizontal sections of a water body. The κ-ε parametrization of turbulence is
applied (Stepanenko et al., 2016).

**3.1.1 Heat transfer in water body**
The water temperature is calculated according to the one-dimensional thermal diffusion
equation:
$c_w \rho_w \frac{\partial T_w}{\partial t} = -c_w \rho_w \frac{1}{A} \int_{\Gamma_A} T_w (u_h \cdot n) dl + \frac{1}{Ah^2} \frac{\partial}{\partial \xi} \left( A K_T \frac{\partial T_w}{\partial \xi} \right) - \frac{1}{Ah} \frac{\partial AS}{\partial \xi} +$
$c_w \rho_w \frac{dh}{dt} \frac{\xi}{h} \frac{\partial T_w}{\partial \xi} + \frac{1}{Ah} \frac{\partial A}{\partial \xi} \left[ S_b(\xi) + F_{iz,b}(\xi) \right]$ ,     (1)
where $c_w$ is heat capacity of water, $\rho_w$ is density of water, $T_w$ is temperature of water,
$h(t)$ is lake depth, $t$ is time, $\xi = z/h$ is a normalized vertical coordinate, $z = 0$ ($z \in$
$[0, h]$) is located at the free surface of the lake, $S$ is downwelling shortwave radiation,
$A$ is the z-dependent cross-sectional area of water, $K_T$ is thermal diffusivity coefficient
equal to the sum of molecular and turbulent diffusivities, $S_b(\xi)$ is shortwave radiation
flux, $F_{iz,b}$ is soil heat flux at the level $z$, $n$ is an outer normal vector to the boundary $\Gamma_A$
of the horizontal cross section $A$ and $u_h$ is horizontal vector in water (Stepanenko et al.,
2016; Guseva et al., 2016). The solar radiation penetrated into the water is calculated
using the Beer-Lambert law (Stepanenko and Lykossov, 2005; Stepanenko et al., 2019):
$S(\xi) = S(0) \exp(-a_e h \xi)$ ,     (2)
where $a_e$ is extinction coefficient. In order to solve the temperature in Eq. (1), it is
necessary to specify the top and bottom boundary conditions and to give the method to
calculate the edge heat flux at each depth $z$. The atmospheric turbulent heat flux
schemes are based on the Monin-Obukhov similarity theory, and the top boundary
condition is a perfect heat balance equation (Stepanenko et al., 2016). When the lake is
covered with ice, the temperature of the last layer of ice and the first layer of layer of
water are equal and fixed to the melting point temperature (Stepanenko et al., 2019),
which is calculated by the following formula:
$T_{mp} = - C * \left| \partial T_{mp} / \partial C \right|$ ,     (3)
where $T_{mp}$ is the melting point temperature (°C), $C$ is salinity at the water-ice interface,
$\left| \partial T_{mp} / \partial C \right| = 66.7$ °C is assumed constant.




### 3.1.2 Heat transfer in snow cover

Snow cover is formed by accumulation of precipitation during the cold season. It is characterized by liquid water content and temperature. The equations are as follows:

$$c_{sn}\rho_{sn}\frac{\partial T_{sn}}{\partial t} = \frac{\partial}{\partial z}\lambda_{sn}\frac{\partial T_{sn}}{\partial z} + \rho_{sn}LF_{fr} - \frac{\partial S}{\partial z} , \qquad (4)$$

$$\frac{\partial W}{\partial t} = -\frac{\partial \gamma}{\partial z} - F_{fr} , \qquad (5)$$

where $c_{sn}$ is specific heat of snow, $\rho_{sn}$ is density of snow, $T_{sn}$ is temperature of snow, $\lambda_{sn}$ is thermal diffusivity of snow, $L$ is latent heat of melting, $F_{fr}$ is rate of freezing, $W$ is liquid water content and $\gamma$ is filtration flux of liquid water (Stepanenko and Lykossov, 2005).

### 3.1.3 Heat transfer in ice cover

The heat conduction equation in ice cover follows the equation:

$$c_i\rho_i\frac{\partial T_i}{\partial t} = c_i\rho_i\frac{\xi}{h_i}\frac{dh_i}{dt}\frac{\partial T_i}{\partial \xi} - c_i\rho_i\frac{1}{h_i}\frac{dh_{i0}}{dt}\frac{\partial T_i}{\partial \xi} - \frac{1}{h_i}\frac{\partial S}{\partial \xi} + \frac{1}{A_ih_i{}^2}\frac{\partial}{\partial \xi}\left(A_i\lambda_i\frac{\partial T_i}{\partial \xi}\right) + \frac{1}{A_ih_i}\frac{\partial A_i}{\partial \xi}F_{T,b} -$$

$$L\rho_i\frac{dp}{dt} , \qquad (6)$$

where $c_i$ is specific heat of ice, $\rho_i$ density of ice, $T_i$ is temperature of ice, $\lambda_i$ is thermal conductivity of ice, $h_i$ is ice thickness, $\frac{dh_{i0}}{dt}$ is the increment of ice thickness on its surface, $F_{T,b}$ is the heat flux at the ice-sediment boundary, $A_i$ is the z-dependent cross-sectional area of the ice cover determined by the basin morphometry, $L$ is the freezing/thawing heat of water and $p$ is ice porosity (Stepanenko et al., 2019).

Based on the study of Leppäranta (2014), the albedo regulates the surface energy budget, and the extinction coefficient controls the vertical distribution of radiation energy in the medium. In the LAKE model, the albedo of water (Aw) is 0.06, and the extinction coefficient of snow (Es) decreases with the increase of snow density. Snow accumulation in the Ngoring Lake area is basically zero, and therefore, only the Ai, the Ei and Ew are analyzed in this study. The model used in this article is version 2.3 of the LAKE model, called LAKE2.3.

This model has been widely used. Thiery et al. (2014) compared and evaluated seven models in LakeMIP by using Kivu Lake, one of the five Great Lakes in Africa. It was found that the LAKE model can better simulate the vertical mixing process and internal thermal stratification of Kivu Lake than Flake and Hostetler models. Stepanenko et al. (2016) found that the LAKE model can reproduce the temperature of Kuivajärvi Lake and the vertical distribution of dissolved gases in summer.





**3.2 Validation Methods**

The indexes to evaluate the accuracy of the model are the root mean square error
($RMSE$), $BIAS,$ and correlation coefficient ($CC$):

$$RMSE = \sqrt{\frac{1}{n}\sum_{i=0}^{n}(m_i - o_i)^2} \ , \qquad (7)$$

$$BIAS = \bar{m} - \bar{o} \ , \qquad (8)$$

$$CC = \frac{\mathrm{Cov}(M,O)}{\sqrt{\mathrm{Var}(M)\,\mathrm{Var}(O)}} \ , \qquad (9)$$

where $m_i$ represents the simulations and $o_i$ represents the observations, $\bar{m}$ is the
average value of simulations and $\bar{o}$ is the average value of observations. $\mathrm{Cov}(M, O)$ is
the covariance of observed and simulated values. $\mathrm{Var}(M)$ and $\mathrm{Var}(O)$ are the variances
of simulated and observed values, respectively.

**3.3 Calculation Method of Heat Storage**

The evolution of the heat storage per unit area in the water is calculated by using the
changes of water temperature profiles

$$\Delta Q = c\rho \sum_{i=1}^{n} T_i \Delta z_i \ , \qquad (10)$$

where c = 4192 J kg$^{-1}$ K$^{-1}$ and $\rho = 10^3$ kg m$^{-3}$, $n$ is the number of depths, $\Delta z_i$ is the depth
interval between two layers and $\Delta T_i$ is the temperature change in layer $i$ (Gan and Liu,
2020; Nordbo et al., 2011).

**4 Characteristics Analysis**

**4.1 Characteristics of Observed Water Temperature**

**4.1.1 Ngoring Lake**

According to the observations in Ngoring Lake, the water temperature has increased
continuously during the winter ice-covered period (Wang et al., 2021; Kirillin et al.,
2021). From the observed water temperature profile (Fig. 2a), it can be seen that the
temperature reached its lowest point in early December 2015, then the lake froze over,
and the water under ice warmed and was completely mixed in the early stage of the ice
season. From mid-March onward, the water body showed weak stratification with
temperature decreasing downward with 5.2 ℃ at 2 m dept and 3.9 ℃ at the bottom. By
mid-April, the ice melted completely, and after that full mixing took place.

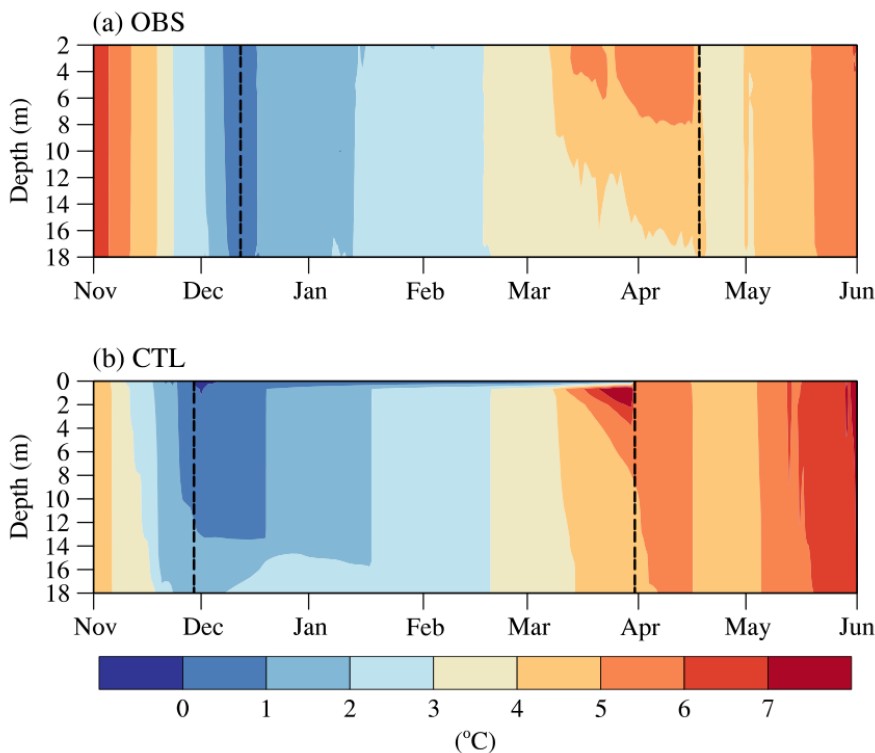

297

**Figure 2. The daily average water temperature (a) observed and (b) simulated in CTL from November 2015 to June 2016. Ice-covered period is represented between the two black dotted lines.**

In order to more intuitively analyze the changes of lake temperature over time, the observed water temperature of Ngoring Lake was averaged daily (Fig. 3a). The lake was mixed completely in early November 2015, and the water temperature decreased gradually. In late November, the temperature oscillated and the difference between 2 m and 22 m was less than 1 ℃. On December 12, the water temperature reached the lowest point, at 2 m it was 0.47 ℃. At this time, the lake was completely frozen, and the air temperature at 2 m height was -7.79 ℃. Thereafter, the lake was mixed, and then the temperature showed new oscillations from early January for about one month (Kirillin et al., 2021). In mid-February, the lake was again fully mixed, and the water temperature continued to rise, reaching $T_{\rho,max}$ on March 7. Thereafter, as the water continued to absorb the strengthening solar radiation, the lake began to stratify, since absorption of radiation decays with depth according to the Beer-Lambert law. The water temperature continued to rise in the upper layer (2-6 m) by the rate ~0.052 ℃ d$^{-1}$, which was higher





than before March 7 (~0.035 °C d$^{-1}$). On April 18, the ice had melted completely, and
the water temperature rose to a maximum of 5.83 °C at 2 m while remaining at $T_{\rho,max}$
below 10 m depth. Having reached the open surface state, full mixing took place rapidly,
and then the lake warmed gaining heat from the atmosphere.

**4.1.2 Kilpisjärvi Lake**

Different from Ngoring Lake, the temperature of some lakes can remain fairly stable in
each layer during ice-covered period, such as Thrush Lake, Valkea-Kotinen Lake,
Pääjärvi Lake and Kilpisjärvi Lake (Fang and Stefan, 1996; Saloranta et al., 2009; Li
et al., 2016b; Tolonen, 1998). Among these lakes, we paid special attention to
Kilpisjärvi Lake.
Kilpisjärvi Lake (K Lake, 69.05º N, 20.83º E, 473 m a.s.l.) is an Arctic tundra lake
located in northern Finland. The latitude difference is about 34°, the longitude
difference is about 77° and the altitude difference is about 3800 m between Ngoring
Lake and K Lake. The temperature decreases with altitude at a rate of 6 °C km$^{-1}$ (Jiang
et al., 2016), and according to present general climatology, in Eastern Europe/Central
Asia the latitudinal and longitudinal decreases in winter are about 1.2 °C deg$^{-1}$ and 0.3 °C
deg$^{-1}$, respectively. Therefore, an increase of 1 km in altitude is equivalent to an increase
of 5° in latitude or 20° in longitude. Between Ngoring and K Lake, the effects of altitude,
latitude and longitude on air temperature offset each other, and there was only little
difference in air temperature between the two sites (Fig. 4f). However, apart from the
surface layer, the water temperature of K Lake basically maintained at $T_{\rho,max}$ during the
ice-covered period (Fig. 3b). Therefore, we suspected that the warming characteristics
of Ngoring Lake were related to the local climate, and to show that we shall analyze
the climate characteristics of the two sites in Sect. 4.2.

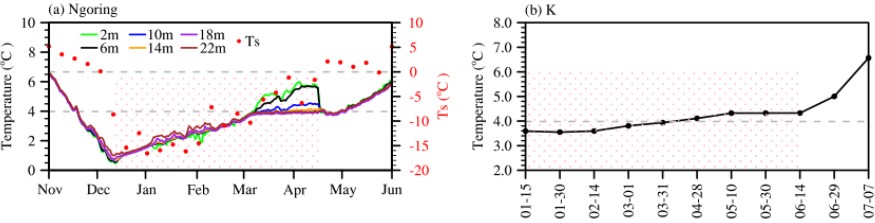

**Figure 3. (a) The daily average water temperature observations of Ngoring Lake**
**at the surface (Ts), 2 m, 6 m, 10 m, 14 m, 18 m and 22 m from November 2015 to**
**June 2016, and (b) the average water temperature observations of K Lake from 5**
**m to 30 m in 1993. Ts is MODIS surface water temperature. The gray reference**
**lines denote $T_{\rho,max}$ = 3.98 °C and 0 °C, respectively. The pink shaded areas denote**
**ice-covered period.**

**4.2 Characteristics of Local Climate**
The daily averages of meteorological variables near the two lakes are shown in Fig. 4
during November to June, and the ranges and averages from December 12 to April 18
of the next year (ice-covered period of Ngoring Lake) were compared (Table 1). The
differences in the average air temperature, specific humidity and downward LR
between Ngoring Lake and K Lake were -0.42 °C, -0.38 g kg$^{-1}$ and 41.9 W m$^{-2}$,
respectively. The wind speed of Ngoring Lake was 1.7 times that of K Lake, and the
downward SR was 159.0 W m$^{-2}$ greater in Ngoring Lake than in K Lake. However, the
precipitation was much less in Ngoring Lake than in K Lake, by the factor of 0.037.
In general, there were not many differences in temperature, specific humidity and
downward LR between the two places, but there was lower precipitation, and higher
downward SR and wind speed in the TP. Since the surface pressure has little effect on
water temperature, this paper does not consider that.

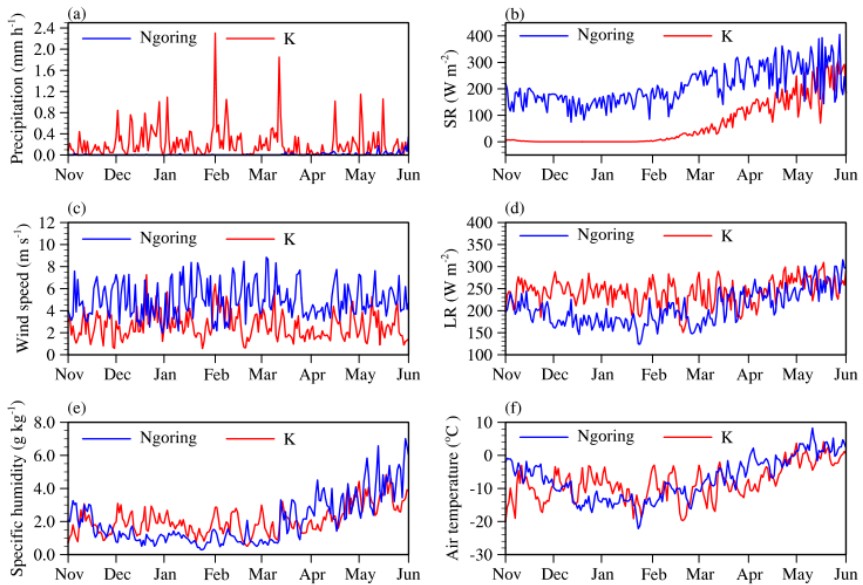

**Figure 4. Comparison of daily average values of the climate field for Ngoring Lake**
**from November 2015 to June 2016 and for K Lake with the same month from 1992**
**to 1993. (a) precipitation, (b) downward SR, (c) wind speed at 10 m height, (d)**
**downward LR, (e) specific humidity and (f) air temperature at 2 m height. The**
**driving field of K Lake was extracted from ERA5-Land data.**





**Table 1.** The range and mean values of the driving meteorological variables of Ngoring
Lake (2015-2016) compared with K Lake (1992-1993) during the ice-covered period
367  (12.12-4.18).

| Meteorologic variables | Ngoring Lake | | K Lake | |
|---|---|---|---|---|
| | Range | Average | Range | Average |
| Precipitation (mm h$^{-1}$) | < 0.072 | 0.0044 | < 1.15 | 0.12 |
| Downward SR (W m$^{-2}$) | 73.98-356.29 | 199.41 | < 186.84 | 40.46 |
| Wind speed (m s$^{-1}$) | 1.95-8.85 | 4.93 | 0.56-7.28 | 2.83 |
| Downward LR (W m$^{-2}$) | 123.92-271.60 | 191.73 | 150.61-289.59 | 233.62 |
| Specific humidity (g kg$^{-1}$) | 0.29-4.52 | 1.40 | 0.50-3.23 | 1.78 |
| Air temperature (°C) | -22.16-2.24 | -10.25 | -19.69--2.01 | -9.83 |


## 5 Simulation Setup

In order to reveal the mechanism of water temperature increase during ice-covered
period in Ngoring Lake and its influences, we set up one control simulation (CTL) and
28 experimental simulations (SIM) in this study (Table 2).

### 5.1 Setup in CTL

The depth of Ngoring Lake was set as 26.5 m, which is the depth at the water
temperature site, and the vertical stratification was described by 35 layers. The
simulation period was from September 2015 to September 2016. The initial vertical
profile of water temperature, and the mixed layer and the bottom temperature were set
according to the observations (Fig. 2a). The albedo of snow and ice and the extinction





coefficients of ice and water were set as, respectively, $A_s = 0.7$, $A_i = 0.25$, $E_i = 2.5$ m$^{-1}$
and $E_w = 0.15$ m$^{-1}$ on the basis of previous investigations (Lei et al., 2011; Li et al.,
2018; Li et al., 2020; Shang et al., 2018). The driving meteorological input variables
were air pressure, wind speed, specific humidity, air temperature, precipitation,
downward SR and LR. The driving data time step was 30 minutes, and the model time
step was 15 seconds.

**5.2 Setup in SIM**
In order to explore the influences of climate field, the simulation called SIM_KinN was
set up where all the forcing variables were replaced by those of K Lake on the basis of
CTL. In order to explore the influence of a single meteorological variable, SIM_*
simulations (* is SR, Precip, LR, U, Tair or q) were set up, where the * variable was
replaced by the K Lake variable on the basis of CTL. These scenarios were quite
artificial because climate variables are actually closely correlated. Nevertheless, using
the sensitivity simulations can shed light on the influence of climate on lake temperature
evolution during ice-covered period.
In order to explore the influence of main physical parameters, SIM_# (# is the value of
$A_i$, $E_i$ or $E_w$) is set up, representing the simulation when $A_i$, $E_i$ or $E_w$ was equal to # on
the basis of CTL, respectively.
SIM_E* (* represents 1, 2 or 3) is set for exploring the effects of different water
temperature profiles before ice-melting on the lake heat storage and heat fluxes, which
represents three different vertical lake water temperature profiles on March 25, 5 days
before the ice breaking.

**Table 2.** Names, explanations and corresponding numbers of all experiments.

| Experiment name | Experiment introduction | Number |
| --- | --- | --- |
| CTL | Control simulation | 1 |
| SIM_KinN | The simulation when all the drive variables are replaced by those of K Lake on the basis of CTL. | 1 |
| SIM_* <br><br> (* represents meteorological variables) | The simulation when the * variable is replaced by that of K Lake on the basis of CTL. | 6 |
| SIM_# | The simulation when the physical variable is | 18 |





| (# represents values of $A_i$, $E_i$ or $E_w$) | equal to # on the basis of CTL, respectively. | |
|---|---|---|
| SIM_E* (* represents 1, 2, and 3) | The simulation when using three different initial temperature profiles on the basis of CTL. | 3 |


## 6 Simulation Results

### 6.1 Model Validation

Compared with the observations (Fig. 2a), the simulation results of CTL (Fig. 2b) were basically consistent with the observations, but the whole ice season was shifted to occur about half a month earlier than observed. The water temperature was slightly higher from mid-March to the end of May in the model compared with observations, and there were differences in the simulation of deep layers. After the ice had melted, the simulated temperature rose faster and was about 1 ℃ higher than the observed value. In general, LAKE2.3 can reproduce the thermal stratification at the end of ice season in Ngoring Lake.

The simulation was evaluated by comparing *RMSE, BIAS, CC* (Table 3) of the simulated and observed water temperature at lake surface, 2 m, 9 m, 14 m and 22 m in Ngoring Lake from November 2015 to June 2016 (Fig. 5a-e). It can be seen that *CC* of each layer was greater than or equal to 0.95, and the *CC* of 2 m, 9 m and 14 m were as high as 0.98, but *RMSE* and *BIAS* of lake surface were larger, 3.25 ℃ and 1.42 ℃, respectively. The surface temperature error was largely owing to the inaccuracy of the MODIS data (Donlon et al., 2002; Tavares et al., 2019). The absolute value of *BIAS* of other layers was less than 0.01 ℃, and *RMSE* was less than 0.95 ℃. In addition, it can be concluded that the simulation of the temperature rise in the ice-covered period was good, and the maximum temperature was 0-1 ℃ higher than the observed value.

In conclusion, LAKE model can simulate the lake warming phenomenon under ice cover good in Ngoring Lake, because in the presence of ice and snow cover, the water temperature of the first layer (0 m) is determined by the freezing point (Eq. 3) and does not depend on air–lake heat exchange, and the solar energy is transferred from the first layer to ice bottom or to the second layer. With the increase of depth, the solar energy absorption decays. Thus, the second layer gains the most of solar heating, while the deeper water temperature maintains at $T_{\rho,max}$. The upper layer is less dense, the stratification is stable, and convection does not occur.

434

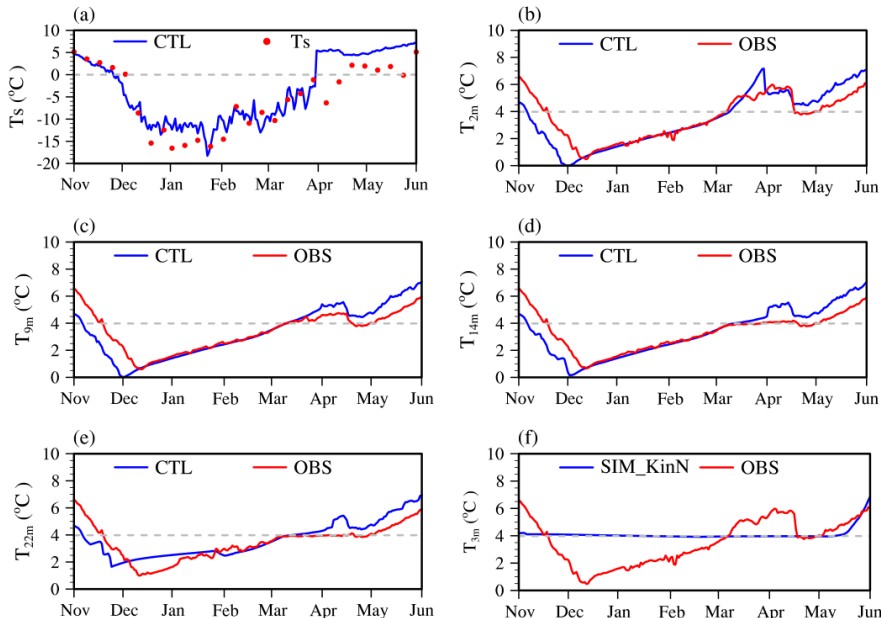

**Figure 5. The daily average water temperature observed and simulated in CTL of (a) the surface (Ts), (b) 2 m, (c) 9 m, (d) 14 m and (e) 22 m in Ngoring Lake from November 2015 to June 2016. (f) Comparison of simulated in SIM_NinK and observed 3 m water temperature in Ngoring Lake. The dotted line represents $T_{\rho,max}$ = 3.98 °C.**

**Table 3.** *BIAS*, *RMSE* and *CC* between simulation and observation corresponded to Fig. 5.

|  | Ts | $T_{2m}$ | $T_{9m}$ | $T_{14m}$ | $T_{22m}$ |
|---|---|---|---|---|---|
| *BIAS* (°C) | 1.42 | 0.04 | 0.09 | -0.05 | 0.06 |
| *RMSE* (°C) | 3.25 | 0.92 | 0.89 | 0.87 | 0.89 |
| *CC* | 0.96 | 0.98 | 0.98 | 0.98 | 0.95 |

**6.2 Influences of Local Climate on Water Temperature**

In order to explore the influences of local climate characteristics on the warming, water temperature and stratification of lakes, we designed 7 simulations, namely, SIM_* (* represents 6 meteorological variables) and SIM_KinN (Table 1). Since the water

temperature at different depths changed consistently with time, the water temperature
at 3 m was selected to analyze the simulation results.
SIM_SR was the simulation of Ngoring Lake when the downward SR of K Lake was
substituted for the forcing. During the ice-covered period, the downward SR of CTL
was strong, with an average of 199.41 W m$^{-2}$, while SIM_SR was 40.46 W m$^{-2}$, with a
difference of 158.95 W m$^{-2}$ with CTL. In the sensitivity experiment SIM_SR, the water
temperature of 3 m was stable keeping in the range of 0-0.1 °C (Fig. 6a). The date of
ice formation was earlier and the melting date was delayed, which led to the growth of
the whole ice-covered period. Compared with CTL, the depth of the mixed layer
increased (Fig. 6d). Thus, during the ice period, the strong downward SR on the TP
caused the water temperature to rise, because SR in Ngoring Lake transferred more heat
through the ice, resulting in the accumulation of heat and continuous warming of the
lake.
In SIM_Precip simulation, the precipitation of K Lake was substituted for Ngoring Lake.
In the sensitivity experiment SIM_Precip, the water temperature at 3 m kept horizontal
in the early and then increased but did not exceed $T_{\rho,max}$ (Fig. 6a). Compared with CTL
(Fig. 2b), the layering and the temperature maximum centers between March and April
disappeared, and the lake was fully mixed (Fig. 6g). This was because the mean
precipitation of SIM_Precip (0.12 mm h$^{-1}$) was approximately 30 times higher than in
the CTL (0.004 mm h$^{-1}$) during the ice-covered period. The increase in precipitation led
to more snowfall, more radiation reflected and absorbed by snow and less radiation
entering water. More precipitation damped the rise in water temperature.
In SIM_LR simulation, the downward LR of K Lake was substituted for Ngoring Lake.
In the ice-covered period, the average downward LR of SIM_LR was 233.62 W m$^{-2}$,
which was larger than in CTL (191.73 W m$^{-2}$). In SIM_LR, the water temperature at 3
m still kept rising, and the time of complete melting of ice was earlier than in CTL, in
the end of February or early March. After the ice breakup, the air temperature was lower,
and the lake transferred heat to the atmosphere, and water temperature underwent a
cooling process (2 °C) until reaching a new equilibrium with the atmosphere (Fig. 6b).
Compared with the CTL, mixing in the ice-covered period was more uniform, the
stratification between March and April was weakened, and the temperature maximum
center was about 15 days earlier (Fig. 6e).
In SIM_U simulation, the wind speed of K Lake was substituted for Ngoring Lake. The
wind speed of SIM_U was less than in CTL for the whole simulation period, and the
average wind speed in ice-covered period was 2.21 m s$^{-1}$, smaller than in CTL. In the
sensitivity experiment SIM_U, the water temperature at 3 m was rising, but it was about
3 °C higher than in the CTL in the whole simulation period (Fig. 6b). Due to the
decrease of wind speed, the depth of mixed layer decreased, and the stability of the lake



stratification increased (Fig. 6h).
In SIM_$T_{air}$ simulation, the air temperature of K Lake was substituted for Ngoring Lake.
The average air temperature difference between SIM_$T_{air}$ (-9.83 °C) and CTL (-
10.25 °C) was small, especially during the ice-covered period, about 0.42 °C. In the
sensitivity experiment SIM_$T_{air}$, the water temperature decreased faster than in CTL,
and at the end of October, the lake began to freeze no longer releasing energy to the
atmosphere. The air temperature of K Lake fluctuated while the temperature of Ngoring
Lake was continuously decreasing (Fig. 6f). The lake stratification was enhanced, and
the maximum center of water temperature was about 10 days ahead of time (Fig. 6f).
In SIM_q simulation, the specific humidity of K Lake was substituted for Ngoring Lake.
The difference of specific humidity between SIM_q and CTL were 0.38 g kg$^{-1}$ during
the ice-covered period. In the sensitivity experiment SIM_q, the simulation results were
similar to the CTL, and thus the specific humidity had little effect on the water
temperature (Figs. 6c and 6i).
On the whole, the stronger downward SR and lower precipitation in the TP played a
positive role to increase the water temperature during the ice-covered period in Ngoring
Lake. Less downward LR, lower air temperature and larger wind speed has an opposite
effect, and specific humidity had no significant influence.

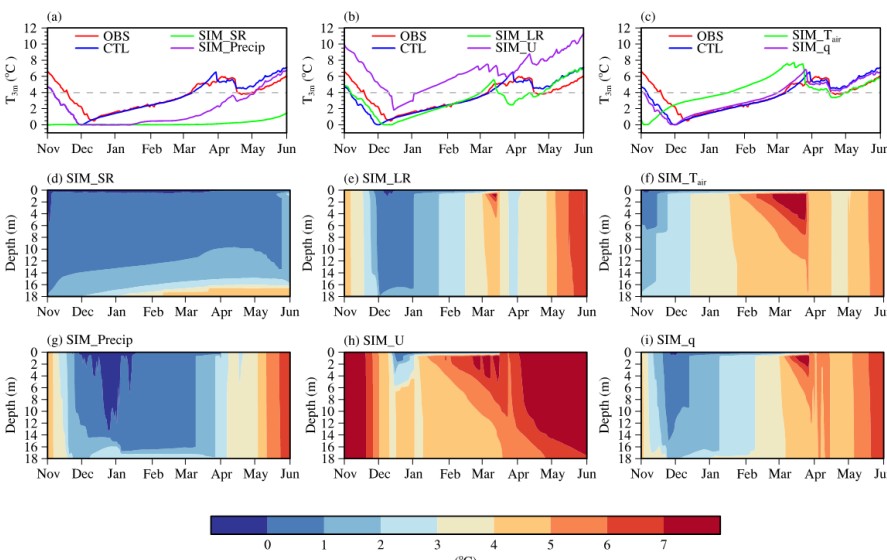

**Figure 6. The simulated 3 m daily average water temperature in (a) (d) SIM_SR,**
**(a) (g) SIM_Precip, (b) (e) SIM_LR, (b) (h) SIM_U, (c) (f) SIM_$T_{air}$, (c) (i) SIM_q**
**experiments from November 2015 to June 2016 are compared with the CTL and**
**the observation, and the change of vertical stratification is shown. The dotted line**

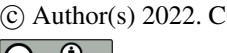



represents $T_{\rho,max} = 3.98\ °C$.

**6.3 Influences of Main Physical Parameters on Water Temperature**
The radiation transfer, depending on the albedo and extinction coefficient, plays a
decisive role on the water temperature. The Ngoring Lake has less snow, and therefore
the influence of $A_i$, $E_i$ and $E_w$ on the lake temperature simulation are discussed with
sensitivity experiments. When the lake is covered by snow, the albedo of dry and light
snow-covered ice is as high as about 0.9 (Leppäranta, 2014; Perovich and Polashenski,
2012). According to previous observations, Ai's observed in the TP were mostly less
than 0.12, and the albedo of clear blue ice was only 0.075 (Li et al., 2018). The range
of $A_i$ without snow cover was set as 0.1-0.8 with an interval of 0.1 in the experiments
SIM_$A_i$. $E_i$ has not been observed on the TP, but surveys in Finnish lakes show that the
value of bare ice varies between 1-4 $m^{-1}$, while the value of snow-covered ice can reach
5 $m^{-1}$ (Lei et al., 2011). In SIM_$E_i$ simulations $E_i$ was equal to 1-5 $m^{-1}$ with an interval
of 1 $m^{-1}$. For the $E_w$, Zolfaghari et al. (2017) found that the FLake model is particularly
sensitive at $E_w \leq 0.5\ m^{-1}$. Shang et al. (2018) observed that $E_i$ varies from 0.11 to 0.67
$m^{-1}$ in a few TP lakes. Therefore, we performed the sensitivity SIM_$E_w$ in which the
$E_w$ varied from 0.1 to 0.5 $m^{-1}$ with an increment step of 0.1 $m^{-1}$. The experimental
settings are shown in Table 4.

**Table 4.** Numerical experimental design of sensitive parameters affecting radiative
transfer.

| Parameter | CTL | SIM_$A_i$ | SIM_$E_i$ | SIM_$E_w$ |
|-----------|-----|-----------|-----------|-----------|
| $A_i$ | 0.25 | 0.1/0.2/0.3/0.4/0.5/0.6/0.7/0.8 | 0.25 | 0.25 |
| $E_i$ ($m^{-1}$) | 2.5 | 2.5 | 1.0/2.0/3.0/4.0/5.0 | 2.5 |
| $E_w$ ($m^{-1}$) | 0.15 | 0.15 | 0.15 | 0.1/0.2/0.3/0.4/0.5 |


In the sensitivity experiment SIM_$A_i$, the water temperature at 3 m decreased with the
increase of ice albedo, which was approximately equal to 1 °C for every step of 0.1 of
the albedos. When the albedo had increased to 0.80, the rise of water temperature had
decreased from 0 °C to 2 °C. The increase of ice albedo does not affect the date of ice



formation, but it delayed the time of ice melting remarkably, thus prolonging the ice-
covered period. When the albedo increased from 0.1 to 0.8, the increase was equivalent
to 0.1-step, and the ice-covered period was extended for 15-30 days (Fig. 7a).
In the sensitivity experiment SIM_$E_i$, changes of extinction coefficient of ice did not
all give a continuous rising of the water temperature, but at 3 m depth the temperature
decreased by 1-2 °C for the increase of ice extinction coefficient by 1 m$^{-1}$ (Fig. 7b). The
greater was the extinction coefficient of ice, the more heat the ice absorbed, and the less
heat entered the lake water under ice.
In order to further explore the influence of $A_i$ on lake temperature in ice-covered period.
We divided ice-covered period into two periods in CTL and the sensitivity experiment
SIM_$A_i$ and SIM_$E_i$, respectively Period-A and Period-B. Period-A ranged from
freezing point to $T_{\rho,max}$, and Period-B ranged from $T_{\rho,max}$ to maximum temperature
(T_max). The duration and heating rate of the two periods and the T_max of Period-B
was calculated (Table 5 & 6). The duration of Period-A is longer than that of Period-B,
and the temperature heating rate of Period-B are 10 orders of magnitude greater than
that of Period-A. The reason is that lake is completely covered by ice, and the inner part
of the lake is evenly mixed in Period-A, while the ice thickness decreases and the
radiation absorbed by the ice decreases in Period-B. The upper layer absorbs more heat
than the deeper layer, and the temperature of the upper layer increases rapidly. When
$A_i$ and $E_i$ increases, the heating rate decreases and the duration increases in Period-A,
the T_max decreases, the heating rate and duration fluctuate in Period-B. When $A_i \geq$
0.6, the heating rate during ice-covered period decreases and will not rise to $T_{\rho,max}$, so
the heating rate and duration of the entire ice-covered period are shown in Table 5.
In the sensitivity experiment SIM_$E_w$, the water extinction coefficient had just little
influence on winter water temperature, which was shown as the late ice temperature
decrease with the increase of $E_w$ (Fig. 7c). The main reason was that in the later period
the ice melted and the ice thickness decreased. The higher was the extinction coefficient
of water, the more heat was absorbed by shallow water and the less heat reached deep
layer.





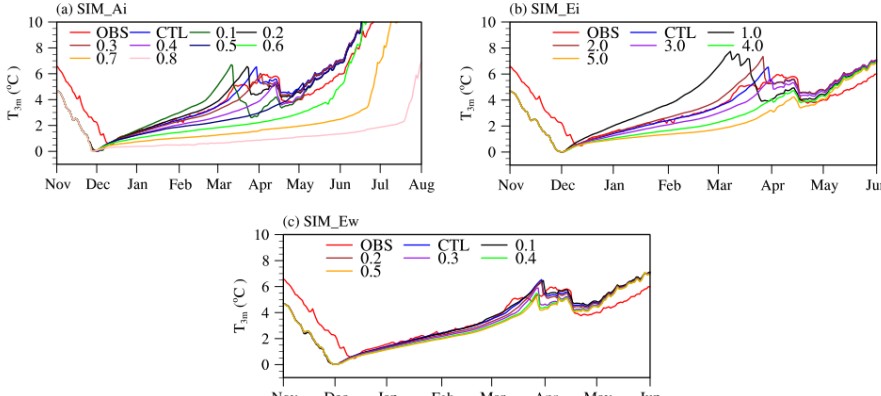

**Figure 7. Comparison of the simulated daily average water temperature of 3 m with the observed value under different (a) $A_i$, (b) $E_i$, (c) $E_w$.**


**Table 5.** The ice-covered period is divided into two periods which Period-A is from freezing point to $T_{\rho,max}$, Period-B is from $T_{\rho,max}$ to T_max. The duration heating rate of two periods and the T_max of Period-B is counted in SIM_$A_i$.

| SIM_$A_i$ | T_max (°C) | Heating Rate (°C d$^{-1}$) | | Duration (day) | |
|---|---|---|---|---|---|
| | Period-B | Period-A | Period-B | Period-A | Period-B |
| 0.1 | 6.70 | 0.048 | 0.144 | 82 | 19 |
| 0.2 | 6.57 | 0.043 | 0.124 | 92 | 21 |
| 0.25 (CTL) | 6.54 | 0.040 | 0.122 | 99 | 21 |
| 0.3 | 6.04 | 0.037 | 0.120 | 106 | 17 |
| 0.4 | 5.36 | 0.032 | 0.127 | 123 | 11 |
| 0.5 | 5.29 | 0.028 | 0.115 | 145 | 11 |
| 0.6 | - | 0.021 | | 175 | |
| 0.7 | - | 0.019 | | 207 | |
| 0.8 | - | 0.011 | | 235 | |

*Here dashes (–) indicate no values.






**Table 6.** The ice-covered period is divided into two periods which Period-A is from
freezing point to $T_{\rho,max}$, Period-B is from $T_{\rho,max}$ to T_max. The duration heating rate of
two periods and the T_max of Period-B is counted in SIM_$E_i$.

| SIM_$E_i$ (m⁻¹) | T_max (°C) | Heating Rate (°C d⁻¹) | | Duration (day) | |
|---|---|---|---|---|---|
| | Period-B | Period-A | Period-B | Period-A | Period-B |
| 1 | 7.53 | 0.059 | 0.101 | 67 | 35 |
| 2 | 7.36 | 0.044 | 0.130 | 91 | 26 |
| 2.5 (CTL) | 6.54 | 0.040 | 0.122 | 99 | 21 |
| 3 | 5.62 | 0.037 | 0.134 | 109 | 12 |
| 4 | 4.73 | 0.033 | 0.052 | 121 | 14 |
| 5 | 4.25 | 0.030 | 0.084 | 132 | 3 |


**6.4 Influences of Water Temperature on Lake-Atmosphere Exchange**
The thermal conditions in an ice-covered lake just before ice melting have significant
influence on the air-lake energy exchange. In order to explore the effects of lake
temperature characteristics on the atmosphere at ice melting, three experiments –
SIM_E1, SIM_E2 and SIM_E3 (Table 1) – were set up based on the CTL and the
observed lake temperature profile on March 25, 2016, 5 days before the ice had
completely melted (Fig. 8a). The characteristics of the initial water temperature profile
were:
- SIM_E1. The stratification was weak, the temperature of the first layer was at
the melting point, and, from the second layer down, the water temperature was
set as 2 °C corresponding to Bangong Co (Wang et al., 2014).
- SIM_E2. The temperature was strongly stratified. The first layer was at the
melting point, and the temperature increased linearly reaching $T_{\rho,max}$ at the
bottom, corresponding to Valkea-Kotinen Lake (Bai et al., 2016).
- SIM_E3. The temperature of the first layer was at the melting point, and the
temperature gradually increased with the depth from the second layer to the
middle layer, and the temperature in the middle layer increased to $T_{\rho,max}$
corresponding to Thrush Lake (Fang and Stefan, 1996).
In the CTL, the first layer was at the melting point, and the second layer reached the
maximum temperature on March 25. The deeper the layer, the lower was the
temperature, until the temperature reached the higher was the temperature until reached
$T_{\rho,max}$.



Under the different initial temperature profiles, the heat storage per unit area of Ngoring
Lake was different after ice breakup, and the difference lasted about two months (Fig.
8b). In CTL, from one day before complete melting (March 30) to complete melting
(March 31), the lake heat content per unit area ranged from 30893.02 MJ m$^{-2}$ to
30874.51 MJ m$^{-2}$, and the heat released was 18.51 MJ m$^{-2}$. In the three experiments, in
the last day before ice complete melting (April 1 to 2), the heat content of the lake
changed from 30657.51 MJ m$^{-2}$ to 30651.67 MJ m$^{-2}$ in SIM_E1, from 30781.07 MJ m$^{-2}$
$^{-2}$ to 30769.91 MJ m$^{-2}$ in SIM_E2, and from 30833.28 MJ m$^{-2}$ to 30822.42 MJ m$^{-2}$ in
SIM_E3, and the heat release was 5.84 MJ m$^{-2}$, 11.16 MJ m$^{-2}$, and 10.86 MJ m$^{-2}$,
respectively (Fig. 8b). Although the initial lake temperature profiles were different
before complete melting, the higher the lake temperature was, the earlier and faster the
ice melted. The heat storage per unit area of Ngoring Lake varied from March 25 to
May 24, and the heat release rate of the lake was different under different circumstances.
After the late May, the heat balance between the lake and the atmosphere was the same,
and so the heat storage per unit area of the lake is basically the same after that.

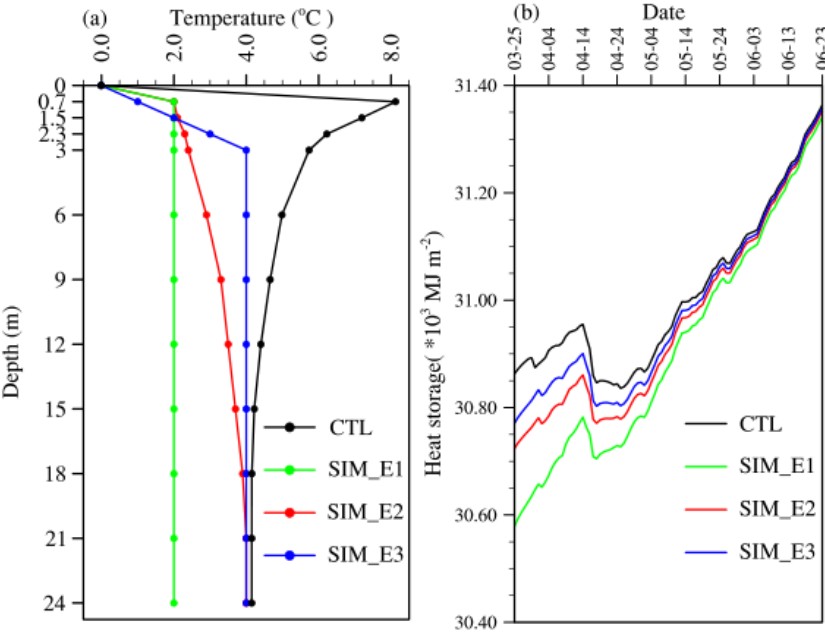

**Figure 8. (a) The initial water temperature profile in the model is set on March 25,**
**2016 and the corresponding daily average (b) lake heat storage per unit area is**
**simulated. SIM_E1, SIM_E2 and SIM_E3 are three different initial water**
**temperature profiles.**




The temperature of the lake surface also affected the sensible and latent heat release
from the lake surface. The sensible and latent heat differences between CTL and the
three experimental simulations were calculated (Fig. 9). The influence of different
initial water temperature profiles started in March 31, that is, when the ice had melted
completely in CTL, and when the sensible and latent differences between CTL and three
experimental simulations was less than 0.1 W m$^{-2}$ for three consecutive days, we judged
that the influence ended. The maximum differences of the sensible heat (51.0 W m$^{-2}$)
and latent heat (76.7 W m$^{-2}$) between SIM_E1 and CTL appeared on March 31 and
ended on June 12 and 30, respectively (Fig. 9a). In SIM_E2 the corresponding numbers
were 51.4 W m$^{-2}$ (March 31 to June 5) for sensible heat and 81.7 W m$^{-2}$ (April 1 to June
17) for latent heat (Fig. 9b), and in SIM_E3 they were 51.5 W m$^{-2}$ (March 31 to May
23) for sensible heat and 86.0 W m$^{-2}$ (April 1 to June 5) for latent heat (Fig. 9c).
Compared with the three lake temperature characteristics, the heating characteristics of
Ngoring Lake made the heat release higher and faster during ice breakup. The duration
of heat release difference was 59 (to May 23)-97 (to June 30) days, and for the latent
heat release the situation lasted about 12-18 days longer than for the sensible heat
release.

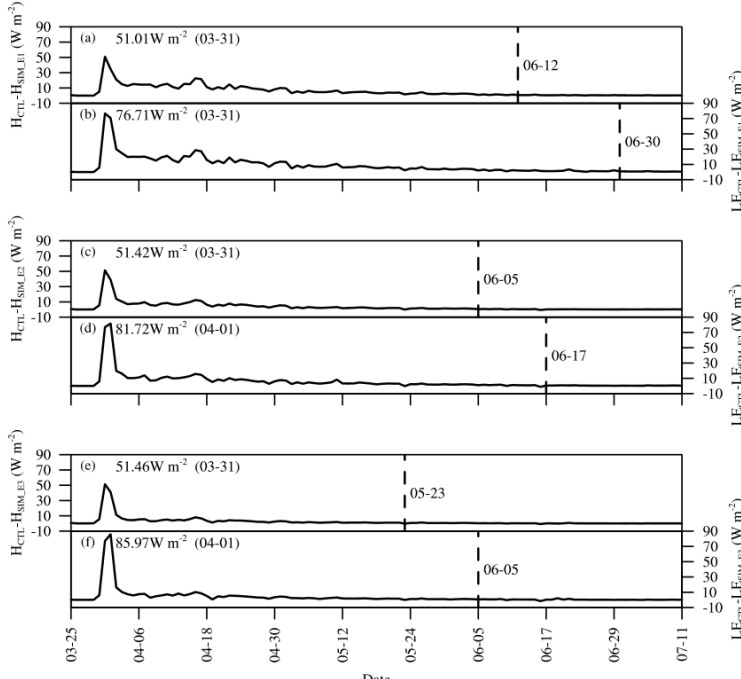

**Figure 9. The daily averaged difference between the simulated sensible and latent**



**heat and the CTL under three different initial water temperature profiles in**
**SIM_E1, SIM_E2 and SIM_E3.**

**7 Conclusions**
In Ngoring Lake, the largest freshwater lake on the TP, we have observed a significant
increase in lake temperature during the ice-period, and this phenomenon not only occurs
in Ngoring Lake but also in other TP lakes such as Bangong Co, Gongzhu Co, Zhari
Namco, Dagze Co and Nam Co. The situation is largely different from the low-altitude
northern lakes where the air temperature is comparable. We used the LAKE model
combined with observed and sensitivity forcing data to study the under-ice water
temperature evolution, revealing the cause, formation mechanism and impact of the
warming phenomenon. The main conclusions are as follows.
During the period from the beginning of freezing to the complete melting of ice, the
water temperature of Ngoring Lake continued to rise. The upper water temperature (2-
10 m) was more than $T_{\rho,max}$, at highest 5.83 °C during 2015 to 2016, while the highest
temperature in deep water was $T_{\rho,max}$.
Different with other tested models (Flake, and the lake scheme coupled in the CLM and
WRF), LAKE2.3 could simulate the vertical thermal stratification during the ice-
covered period in Ngoring Lake well, and the continuous rising of water temperature
was simulated more accurately. Compared with MODIS surface temperature data, the
*BIAS*, *RMSE* and *CC* were 1.42 °C, 3.25 °C and 0.96, respectively. The absolute values
of *BIAS* and *RMSE* were less than 0.1 °C and 1 °C in 2 m, 9 m 14 m and 22 m. The *CC*
of simulated and observed water temperature at 2 m, 9 m and 14 m were as high as 0.98,
and the *CC* of simulated and observed water temperature at 22 m was 0.95.
Sensitivity simulations with perturbed local climate data showed that strong downward
SR in TP played a dominant role in the water temperature rise during the ice-covered
period in Ngoring Lake, and also the low precipitation played a positive feedback role.
The smaller downward LR, lower air temperature and larger wind speed had negative
feedback to the water temperature.
The sensitivity simulation results of the main physical parameters that affect the
radiation transfer showed that with the increase of the albedo of ice, the rising trend of
water temperature decreased and the length of the ice season increased. When albedo
increased to 0.6, the lake water temperature no longer rose but tended to remain on a
stable level. With the increase of extinction coefficient of ice, the increase of the
temperature of the lake in the ice-covered period of Ngoring Lake decreased. The
extinction coefficient of water had just a minor effect on water temperature under ice.
Compared with three more stable lake temperature profiles, the warming of Ngoring



Lake ice-covered period caused the maximum sensible and latent heat releases after ice melting, and the difference of sensible and latent heat relesass lasted for 59-97 days between the lakes with the characteristics of three typical ice-covered periods which the water temperature remained fixed in each layer or was less than or equal to the maximum density temperature and Ngoring Lake. The distribution of water temperature affected the heat storage and heat transfer of lake surface after ice melting. The higher the water temperature, the higher the heat storage per unit area of the lake, and the greater were the sensible and latent heat release from the melting ice.

*Data availability.* The daily precipitation data from Chinese surface stations are available for purchase from the China Meteorological Data Service Center (CMDC, http://data.cma.cn/en/). The MODIS LST product are available from National Aeronautics and Space Administration (NASA) (https://earthdata.nasa.gov/). ERA5-Land data is available with funding from the European Union's Copernicus Climate Change Service (https://cds.climate.copernicus.eu/). Lake temperature data of Ngoring Lake in 2015 and 2016 were uploaded to Zenodo by Georgiy Kirillin (http://doi.org/10.5281/zenodo.4750910). The weather observation data of Ngoring Lake can be obtained from the website (https://nimbus.igb-berlin.de/index.php/s/Moqxgn29DbNFyr8).

*Author contributions.* MW and LW conceived the study. MW performed the modelling with contributions from VS, LW and ZL. YZ, RN and LY processed some data. MW, LW, ML and GK analyzed the model output. MW wrote the paper, with contributions from all co-authors.

*Competing interests.* The authors declare that they have no conflict of interest.

*Acknowledgments.* This study was supported by the National Key Research and Development Program of China (2019YFE0197600) and CAS "Light of West China" Program (E129030101, Y929641001). Victor Stepanenko was supported by Russian Ministry of Science and Higher Education, agreement No. 075-15-2019-1621.

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
