# Peer review of "Mechanisms and effects of under-ice warming water in Ngoring Lake of Qinghai-Tibet Plateau"

_The Cryosphere, 2021_

## Author Comment (AC1)

**Reply to RC1**

We are thankful to the Reviewers for careful reading of the manuscript and helpful comments and suggestions on the study. Both Reviewers presented similar critical points, which we agreed with and followed in the revision of the study. Therefore, we preface the detailed responses with a general description of the main revision points, reflecting the aims of the study, its novelty, and the major results.

Note: the **comments** and **authors' replies** are in font colors of **black** and **blue**, respectively. The **blue content in quotation marks** is the expression marked in red in the revised manuscript. Figure 2 and Figure 3 merged and Figures 4-8 becomes Figure 3-7 in the revised manuscript.

**General response:**

1.  The revised abstract is focused now on the study background, goals, design, and outcomes:

    "The seasonal ice cover in lakes of the Qinghai-Tibet Plateau is a transient and vulnerable part of the cryosphere, whose characteristics depend on the regional climate: strong solar radiation in the context of the dry and cold environment. We use the first under-ice temperature observations from the largest Tibetan freshwater lake Ngoring and a one-dimensional lake model to quantify the mechanism of solar thermal accumulation under ice, which relies on the ice optical properties and weather conditions, as well as the effect of the accumulated heat on the land-atmosphere heat exchange after the ice break-up. The model was able to realistically simulate the feature of Ngoring Lake thermal regime: the "summer-like" temperature stratification with temperatures exceeding the maximum density point of 3.98 ℃ across the bulk of the water column. A series of sensitivity experiments revealed solar radiation was the major source of under-ice warming and demonstrated the warming phenomenon was high sensitivity to the optical properties of ice. The heat accumulated under ice contributed to the heat release from the lake to the atmosphere for 1-2 months after ice-off, increasing the upward sensible and latent surface heat fluxes by ~50 W m$^{-2}$ and ~80 W m$^{-2}$, respectively. Therefore, the delayed effect of heat release on the land-atmosphere interaction requires an adequate representation in regional climate modeling of the Qinghai-

Tibet Plateau and other lake-rich alpine areas."

2. We have deeply revised Introduction and Discussion/Conclusion to better outline the study's aims and its major outcomes. In particular, we state in Introduction that:

"we adopt for this study a "classical" two-equation turbulence modeling approach proving its reliability in decades of studies on the environmental turbulent fluid dynamics. The one-dimensional model LAKE implements the approach in application to lake dynamics and was applied previously to different lakes (Stepanenko et al., 2011, 2016; Guseva et al., 2016). We combine modeling with in situ observations from Ngoring Lake, data on weather forcing and remote sensing to: (i) test the ability of a one-dimensional lake model LAKE to simulate temperature and stratification driven by intense solar heating in ice-covered Lake Ngoring; (ii) conduct series of sensitivity experiments aimed at revealing the role of meteorological forcing and ice optical properties in lake temperature and mixing regime; and (iii) reveal the effects of temperature distribution before ice breakup on lake heat storage and lake- atmosphere heat transfer."

3. As one of the major novel outputs of the study, we mention in Conclusions:

"An important consequence of the under-ice solar heat accumulation consisted in increased sensible and latent heat releases in the subsequent open-water phase. According to the model results, the effects on the surface fluxes of Ngoring Lake lasted for 59-97 days after the ice melt and increased the upward latent and sensible surface heat fluxes up to ~80 W m$^{-2}$ and ~50 W m$^{-2}$, respectively. Herewith, the phenomenon of under-ice solar heating may have a significant effect on the land-atmosphere interaction on regional scales and has to be accounted for in coupled climate models."

4. In the revised study, we focus on the major factors affecting the penetration of solar radiation under lake ice, and later heat release from the water column to the atmosphere after the ice melt.

We hope that these changes have resolved the conceptual issues raised by the Reviewers and provided the study with the right context. Detailed responses and descriptions of changes are given below.

- I think this paper is trying to understand how solar radiation influences thermal stratification under lake ice on a large lake on the Tibetan Plateau. This is an admirable goal, but it isn't clear what the novelty of this paper is. I would say

aspects of this process are well known, and hence introduction needs to better review existing literature and make it clearer what new contribution in this work. I would say it is well known that over winter lakes warm up under ice. This is a key point in the highly cited 2012 review by Kirillin (Who is one of the co-authors on this paper!)

Kirillin, Georgiy, Matti Leppäranta, Arkady Terzhevik, Nikolai Granin, Juliane Bernhardt, Christof Engelhardt, Tatyana Efremova et al. "Physics of seasonally ice-covered lakes: a review." Aquatic sciences 74, no. 4 (2012): 659-682.

Specifically, they introduce idea of Winter I and winter II as periods where heating is dominated by benthic heating (early winter) or solar radiation (late winter). I would say your lake is completely consistent with a long winter II dynamic. Another paper to better review is the GRL paper by Yang et al (2021), who introduce idea of cryomictic and cyrostratified lakes. Based on Figure 4 the lake on TP is windier than lake on Nordic tundra. there is no information on size of Kilpisjärvi Lake, but I assume that is much smaller than 610 km$^2$ Ngoring Lake (which is almost same size as 720 km$^2$ Lake Simcoe). Based on Yang et al (2021) you'd expect Ngoring Lake to be cryomictic and start winter near 0 °C before it warms up, whereas the smaller less windy Kilpisjärvi Lake to start winter nearer 4 °C as a cryostratified lake. I think the novelty of paper needs to be discussed in context of these two papers — This would change statement in abstract about warming dynamics that "The lake water temperature was observed to be generally rising during the ice-covered period from November 2015 to April 2016. This phenomenon appeared in the whole water column, with slowing in deep water and accelerating in shallow water before ice melting. The process is different from low-altitude boreal lakes. There are few studies on its mechanism and effects on lake-atmosphere interaction."

**Reply:** Thanks for your detailed consideration and constructive comments. Your advice improved the total quality of the manuscript. I have read the two articles carefully. We took your comments in deep consideration and revised the manuscript which we hope meet with approval. The **novelty** of this article is mainly reflected in the following three aspects:

1)  In this paper, not only the influence of solar radiation on the warming up of under-ice water temperature was analyzed, but also the influence of other factors

(meteorological conditions and main physical parameters) was first quantitatively for the warming phenomenon.

2) Lakes with large surface area in mid-latitudes, especially those in dry continental climate zones, are snowless in winter. Winter II dominates the entire ice-covered period. During which convection mixing by radiative heating of upper water is dominant (Kirillin et al., 2012).

According to the maximum depth, surface area and wind speed, the lake can be divided into two types which are cryostratific and cryomictic lakes. The premise of two types is that the lake temperature does not exceed 4 ℃ (Yang et al., 2021).

Consistent with lakes of Winter II, solar radiation continued to heat up water during ice-covered period in Ngoring Lake. However, the upper water temperature in lakes of Winer II is less than 4 ℃ due to the heating is not intense enough, which in Ngoring Lake is more than 4 ℃ because of strong solar radiation.

Ngoring Lake mixed evenly when ice formed, that is consistent with cryomictic lakes. However, cryomictic lakes was mixed evenly until melting and the water temperature was less than 4 ℃. While Ngoring Lake was mixed evenly in the early ice-covered stage and over 4 ℃ in the late stage, lake stratified after that.

3) The common lake model CLM-Lake and Flake applied into the Tibetan Plateau could not simulate this phenomenon, so the LAKE model that could do this was introduced into the study. If there was no the suitable lake model for the process simulation, it will hinder the study of the effect of plateau lakes on the atmosphere.

The information about Kilpisjärvi Lake is shown below:

Kilpisjärvi Lake (69.05º N, 20.83º E, 473 m a.s.l.) is an Arctic tundra lake with average depth of 19.5 m and maximum depth of 57 m. The lake has a surface area of 37 km$^2$. It is a cryostratified lake.

The statement in abstract about warming dynamics is unchanged, but the uniqueness is presented. The abstract not repeated here has been revised as the general response.

**Specific comments**

- The section from lines 79 to 104 needs to be completely rewritten. There is no need

to discuss Lake Kivu which is a tropical merimoctic lake. If you want to talk about lake catergorisations, I recommend starting with

Lewis Jr, W. M. (1983). A revised classification of lakes based on mixing. Canadian Journal of Fisheries and Aquatic Sciences, 40(10), 1779-1787.

Then discuss 2012 of Kirillin and 2021 GRL paper by Wang et al. The other papers on winter dynamics of TP need to be better reviewed including

Wang, J., Huang, L., Ju, J., Daut, G., Ma, Q., Zhu, L., Haberzettl, T., Baade, J., Mäusbacher, R., Hamilton, A. and Graves, K., 2020. Seasonal stratification of a deep, high-altitude, dimictic lake: Nam Co, Tibetan Plateau. Journal of Hydrology, 584, p.124668.

**Reply:** Thank you and follow your recommendation. I have deleted the discussion about Kivu Lake and talked about lake categorizations starting with Lewis. Then focused on description of typical processes of under-ice stratification, the differences in lakes on the Qinghai-Tibet Plateau, and application of existing lake models in under-ice water.

The introduction has changed to the following:

[revised manuscript text omitted]

- line 144 - is this lake salty like other TP lakes? this become important later when under ice temps go above 4 °C.

**Reply:** Thanks for your question. Ngoring Lake is a freshwater lake (0.27 g kg$^{-1}$, Shen et al., 2012), which is different from other lakes such as Qinghai Lake (12.3 g kg$^{-1}$), Selin Co Lake (18.7 g kg$^{-1}$), Nam Co Lake (1.7 g kg$^{-1}$) (Wu et al., 2021) and Zhari Namco Lake (14.8 g kg$^{-1}$), Dagze Co Lake (18 g kg$^{-1}$) (Lazhu et al., 2021). According to the simulation, lake temperature can exceed 4 °C without salinity, so salinity does not play an important role.

- Figure 1. Where is Nordic lake?

**Reply:** Thanks for your question. The Nordic lake in the original text refers to Kilpisjärvi Lake (69.05º N, 20.83º E, 473 m a.s.l.), which is located in northern Finland (Lei et al., 2012). The main research area of this paper is Ngoring Lake. Kilpisjärvi Lake provides its driving data as an auxiliary lake different from Ngoring Lake.

● What is bathymetry of lake - we more interested in that than topography. Where is water temperature sampled?

**Reply:** Thanks for your question. Ngoring Lake has a maximum depth of 32 m and an average depth of 17 m. Water temperature is measured at the WS point (35.03º N, 97.70º E, Figure 1). The bathymetry of Ngoring Lake and WS has been added in Figure 1.

[Figure]

**Figure 1. (a) Location of Ngoring Lake, the pentagram denotes the lake border station (LBS) and water temperature measurement point (WS). (b) The bathymetry of Ngoring Lake. (b) adapted from (Kirillin et al., 2021).**

● line 168 - need to say specifically where profile was taken and add to figure 1.

**Reply:** Thanks for your advice. The water temperature measurement site (WS, 35.03º N, 97.70º E) has been added in Figure 1.

● Fgure 2 --Use a continuous shading, not something with 1 ℃ steps, when whole range of interest is really 0 - 4 ℃

**Reply:** Thanks for your suggestion. Figure 2 has been revised with a continuous shading and 0.1 ℃ interval.

[Figure]

**Figure 2. (a) The daily average water temperature observations of Ngoring Lake at the surface (Ts), 2 m, 9 m, and 22 m from November 1, 2015 to June 1, 2016. Ts is MODIS lake surface temperature. The gray reference lines denote 3.98 °C and 0 °C, respectively. The pink shaded area denotes ice-covered period. The water temperature profile (b) observed and (c) simulated in CTL. The ice-covered period is represented between the two red dashed lines.**

- Line 308 - " Thereafter, the lake was mixed,.." You need a discussion in intro about Winter II and solar driven convection for this statement to make sense.

**Reply:** Thank you and follow your suggestion. I discussed the convection caused by solar radiation heating shallow water during ice-covered period. The following sentence has been added in Section 4.1.1:

"Ngoring Lake is mostly covered only by bare ice in winter due to drought, less precipitation and snow. In the early ice-covered phase (from December 12 to March 7), the whole lake mixed completely because solar radiation penetrated ice and heated the

upper water, which was warm ($< T_{md}$), heavy and sinking (Fig. 2b) (Kirillin et al., 2012). In parallel, water temperature continued to warm until reached $T_{md}$ on March 7 (Fig. 2a)."

- Line 329 - Don't abbreviate Kilpisjärvi Lake as K lake. It might be better to refer to it as lake Kilpisjärvi, as jarvi just means lake in Finnish. There are also no details on this lake - how deep how wide? Other publications on this data.

**Reply:** Thanks for your suggestion and the explanation about järvi. All abbreviations K Lake have been changed to Kilpisjärvi Lake in the revised manuscript and K changed to Kilpis in Figure 3. The details of Kilpisjärvi Lake (37 km$^2$, average depth 19.5 m, maximum depth 57 m) as described below have been added in Section 2.2.3:

"ERA5-Land data is applied for a comparative analysis of warming mechanisms and thermal conditions in Tibetan ice-covered lakes against those in the Arctic. The reanalysis forcing data for the geographical position 69.05º N, 20.83º E was adopted as "typical" arctic weather conditions. Northern Fennoscandia is covered by several lakes characterized by the longest ice-covered period in Western Europe. The largest of these lakes, Kilpisjärvi, has a similar morphometrical feature to Ngoring (average depth 19.5 m, maximum depth 57 m, surface area 37 km$^2$). The lake has been intensively studied in the last decades (Kirillin et al., 2015, 2018; Leppäranta et al., 2017, 2019). Its under-ice water temperature remained stable during winter from 1992 to 1993 (Tolonen, 1998). In the following, model experiments forced by the ERA5 weather data (1992-1993) for the Arctic refer to "Kilpisjärvi" runs."

- Figure 3 - use same x-axis formats for dates. Different data for temps is plotted so also hard to compare Y-axis of a and b.

**Reply:** Thanks for your suggestion. According to another reviewer's suggestion that deleting unimportant parts because the manuscript is too long, I have deleted the right panel of Figure 3 and merged the left panel with Figure 2.

- Figure 4 - comment on differences in wind speeds in driving one lake to be cyrostratified and the other cyromictic. The long polar night above artic circle

drives Fig 4 b, so timing of magnitude of solar radiations drives most of differences in under ice convection.

**Reply:** Yes, I agree with you. Compared with Ngoring Lake, the wind speed of Kilpisjärvi Lake is relatively small, so Kilpisjärvi Lake is cryostratified lake which is deeper lakes or those with calmer winds, result in ice forming just above deeper waters of 3–4 °C (Tolonen et al., 1998). Due to the polar night phenomenon during Winter in Kilpisjärvi Lake, the variation range of lake temperature is small and remains stable for a long time.

- Line 370 - this question on under ice heating needs to better motivated by a revised introduction.

**Reply:** Thanks for your suggestion. The introduction has been revised.

---

## Author Comment (AC2)

**Reply to RC2**

We are thankful to the Reviewers for careful reading of the manuscript and helpful comments and suggestions on the study. Both Reviewers presented similar critical points, which we agreed with and followed in the revision of the study. Therefore, we preface the detailed responses with a general description of the main revision points, reflecting the aims of the study, its novelty, and the major results.

Note: the **comments** and **authors' replies** are in font colors of **black** and **blue**, respectively. The **blue content in quotation marks** is the expression marked in red in the revised manuscript. Three supplement figures were added to facilitate understanding of the response, but did not appear in the manuscript. Figure 2 and Figure 3 merged and Figures 4-8 becomes Figure 3-7 in the revised manuscript.

**General response:**

1. The revised abstract is focused now on the study background, goals, design, and outcomes:

   "The seasonal ice cover in lakes of the Qinghai-Tibet Plateau is a transient and vulnerable part of the cryosphere, whose characteristics depend on the regional climate: strong solar radiation in the context of the dry and cold environment. We use the first under-ice temperature observations from the largest Tibetan freshwater lake Ngoring and a one-dimensional lake model to quantify the mechanism of solar thermal accumulation under ice, which relies on the ice optical properties and weather conditions, as well as the effect of the accumulated heat on the land-atmosphere heat exchange after the ice break-up. The model was able to realistically simulate the feature of Ngoring Lake thermal regime: the "summer-like" temperature stratification with temperatures exceeding the maximum density point of 3.98 °C across the bulk of the water column. A series of sensitivity experiments revealed solar radiation was the major source of under-ice warming and demonstrated the warming phenomenon was high sensitivity to the optical properties of ice. The heat accumulated under ice contributed to the heat release from the lake to the atmosphere for 1-2 months after ice-off, increasing the upward sensible and latent surface heat fluxes by $\sim50$ W m$^{-2}$ and $\sim80$ W m$^{-2}$, respectively. Therefore, the delayed effect of heat release on the land-atmosphere interaction

requires an adequate representation in regional climate modeling of the Qinghai-Tibet Plateau and other lake-rich alpine areas."

2.  We have deeply revised Introduction and Discussion/Conclusion to better outline the study's aims and its major outcomes. In particular, we state in Introduction that:

    "we adopt for this study a "classical" two-equation turbulence modeling approach proving its reliability in decades of studies on the environmental turbulent fluid dynamics. The one-dimensional model LAKE implements the approach in application to lake dynamics and was applied previously to different lakes (Stepanenko et al., 2011, 2016; Guseva et al., 2016). We combine modeling with in situ observations from Ngoring Lake, data on weather forcing and remote sensing to: (i) test the ability of a one-dimensional lake model LAKE to simulate temperature and stratification driven by intense solar heating in ice-covered Lake Ngoring; (ii) conduct series of sensitivity experiments aimed at revealing the role of meteorological forcing and ice optical properties in lake temperature and mixing regime; and (iii) reveal the effects of temperature distribution before ice breakup on lake heat storage and lake- atmosphere heat transfer."

3.  As one of the major novel outputs of the study, we mention in Conclusions:

    "An important consequence of the under-ice solar heat accumulation consisted in increased sensible and latent heat releases in the subsequent open-water phase. According to the model results, the effects on the surface fluxes of Ngoring Lake lasted for 59-97 days after the ice melt and increased the upward latent and sensible surface heat fluxes up to ~80 W m$^{-2}$ and ~50 W m$^{-2}$, respectively. Herewith, the phenomenon of under-ice solar heating may have a significant effect on the land-atmosphere interaction on regional scales and has to be accounted for in coupled climate models."

4.  In the revised study, we focus on the major factors affecting the penetration of solar radiation under lake ice, and later heat release from the water column to the atmosphere after the ice melt.

We hope that these changes have resolved the conceptual issues raised by the Reviewers and provided the study with the right context. Detailed responses and descriptions of changes are given below.

- The manuscript (TC-2021-398) by Wang et al. conducted a series of modeling experiments on the formation and development of water temperature rise during

the ice-covered seasons in a large freshwater lake of the Qinghai-Tibet Plateau, where lake ice processes and ice-covered lakes were less studied. It is very interesting that weather forcing data in Lake K were used to run the LAKE to simulate the water temperature regime in Lake Ngoring in order to investigate what causes the differences in lake temperature regimes. The paper concluded the intensive solar radiation and absence of snow dominate the development of temperature rising and the temperature rising has significant influence on after-ice-off lake-air heat exchange.

I found this research topics fits well the scope of TC journal. The methods and data analysis are sound. But I have to say the water temperature rising/stratification dynamics is not new since some published observations on QTP lakes and review papers presented this phenomenon, its mechanism is relatively clear. But we still need validation and evaluation of current lake models to reproduce this process in lakes with large solar energy input. However, the motivation and novelty of this work are not clear, maybe largely due to its poor writing and linguistic expression. The manuscript is too long and should be more concise and focused by removing less-related and repeated parts. So, I think plenty of extra work is needed to improve the overall quality of this manuscript before it can be considered for publication. Please see my comments below and I hope they would be useful for authors' consideration.

**Reply:** Thanks for your deep consideration, constructive advice and affirmation. These comments are all valuable and very helpful for revising and improving our paper, as well as the important guiding significance to our researches. We have studied comments carefully and have made correction which we hope meet with approval.

1. We still need validation and evaluation of current lake models to reproduce this process in lakes with large solar energy input. Due to the difficulty of observation, there are few studies on lake ice processes and ice-covered lakes in Qinghai-Tibet Plateau. Lake models are often used to simulate their processes and reveal physical mechanisms. However, the common models such as CLM-Lake and Flake cannot accurately simulate the plateau lakes temperature characteristics during ice-covered period. Therefore, a new model called LAKE, was introduced to evaluate its applicability in the Ngoring Lake during ice-covered period. Then the model can be used to extensively simulate plateau lakes to reveal their physical mechanisms. In addition, the LAKE model contains ecological modules, which can

simulate the ecological process inside lake. The LAKE model can also be coupled with the atmospheric model to reveal the interaction between lakes and atmosphere. Therefore, it is important to validate and evaluate existing lake models.

2. The characteristics of lake temperature during ice-covered period of Ngoring Lake are unique and different from those of existing researches. The motivation and innovation may not be clearly expressed due to language expression, which has been further stated in the revised draft. Its innovation is mainly reflected in the following three aspects:

   a) In this paper, not only the influence of solar radiation on the warming up of under-ice water temperature was analyzed, but also the influence of other factors (meteorological conditions and main physical parameters) was first quantitatively for the warming phenomenon.

   b) Lakes with large surface area in mid-latitudes, especially those in dry continental climate zones, are snowless in winter. Winter II dominates the entire ice-covered period. During which convection mixing by radiative heating of upper water is dominant (Kirillin et al., 2012).

   According to the maximum depth, surface area and wind speed, the lake can be divided into two types which are cryostratific and cryomictic lakes. The premise of two types is that the lake temperature does not exceed 4 °C (Yang et al., 2021).

   Consistent with lakes of Winter II, solar radiation continued to heat up water during ice-covered period in Ngoring Lake. However, the upper water temperature in lakes of Winer II is less than 4 °C due to the heating is not intense enough, which in Ngoring Lake is more than 4 °C in because of strong solar radiation.

   Ngoring Lake mixed evenly when ice formed, that is consistent with cryomictic lakes. However, cryomictic lakes was mixed evenly until melting and the water temperature was less than 4 °C. While Ngoring Lake was mixed evenly in the early ice-covered stage and over 4 °C in the late stage, lake stratified after that.

   c) The common lake model CLM-Lake and Flake applied into the Tibetan Plateau could not simulate this phenomenon, so the LAKE model that could

do this was introduced into the study. If there was no the suitable lake model for the process simulation, it will hinder the study of the effect of plateau lakes on the atmosphere.

3. According to the suggestion that the manuscript is too long and should be more concise and focused, Section 3.1.2 heat transfer in snow is deleted since snow is not involved in the study. Table 3, 4 and 5 have been deleted.

**General issues:**

- The manuscript carried out many sensitivity experiments to varied forcing data and coefficients. But physically, from the perspective of heat budget and balance of the lake water, only the solar radiation and snow condition can directly change the water temperature, other weather variables have indirect impacts on under-ice water, such as wind, air temperature, and air humidity. I mean these meteor variables directly influence the heat and mass balance of the ice cover.

  So, one question is, can the LAKE model give the lake ice thickness and temperature evolution? If you look at the modeling results on lake ice evolution under all modelling experiments, it would be easier to understand why the meteor variables and coefficients have differing impacts on water temperature under the ice (like sections 6.2 and 6.3).

**Reply:** Thanks for your question. I quite agree with you that solar radiation and snow cover change the water temperature directly. Other variables directly affect the heat and mass balance of ice by changing the ice thickness and thus influence the lake temperature indirectly. For example, in SIM_$A_i$ experiments, the ice-covered period and ice thickness increase as the ice albedo increases. which leads to less radiation entering the water and lower lake temperatures.

The LAKE model can simulate the evolution of ice thickness and temperature. Ice thickness and temperature in CTL are shown in Supplement: Figure 1. The ice is divided into five layers on average, The ice temperature at first layer is equal to the surface temperature, at the last layer is equal to ice water mixture temperature (Supplement: Figure 1a).

[Figure]

[Figure]

**Supplement: Figure 1. The daily average (a) ice temperature and (b) thickness simulated in CTL. Tsurf_CTL and Tsurf_modis represent simulated surface temperature and MODIS surface temperature. Tice_layer 1, 2, 5 represent simulated ice temperature in layer 1, 2 and 5.**

● Another question is, when the water temperature goes beyond the temperature of maximum density, a temperature dichotomy forms (as presented in the GRL paper of the authors, Kirillin et al, 2021, Ice-covered lakes of Tibetan plateau as solar heat collectors, GRL). And the regime of the dichotomy layer is of great importance to the water temperature and heat storage, but the inverse temperature gradient/structure above the temperature peak point seems unstable since the temperature crosses the temperature of maximum density. Do you have salinity profiles to look into this regime? Or did the LAKE model reproduce the dichotomy well? How does it change when the solar radiation, optical coefficients of ice and water change? Could you discuss on this? This is important to evaluate the performance of LAKE to reproduce the temperature structure beside value.

**Reply:** Thanks for your question. There is no salinity profile of Ngoring Lake, but based on the study of Shen et al. (2012), as shown in Supplement: Figure 2, the initial salinity profile is set as vertical uniform to 0.27 g kg$^{-1}$.

[Figure]

**Supplement: Figure 2. The vertical salinity distribution in Ngoring Lake (Shen et al., 2012).**

LAKE model reproduced the dichotomy well. The dichotomy changes affected by the radiation and the coefficients change have been discussed based on the experiments using the lake model in the revised manuscript.

When the solar radiation of Ngoring Lake was replaced by that of Kilpisjärvi Lake which is 0 W m$^{-2}$ due to the polar night. The water temperature no longer warmed up during ice-covered period, resulting in stratification and the dichotomy disappeared. When the ice albedo is greater than 0.5, the lake temperature does not rise more than 4 °C, and the dichotomy disappeared. When the extinction coefficient of ice is greater than 4.0 m$^{-1}$, the dichotomy disappeared. The extinction coefficient of water did not make the dichotomy disappear.

**Specific comments:**

- I also recommend the manuscript to be language checked to erase linguistic errors (I found some as below), misunderstandings, and confusions.

**Reply:** Thanks for your suggestion. The language has been improved significantly by a specialized language service in the revised manuscript.

- Abstract: although the abstract is very long, I do not clearly get the key points on what this manuscript did, found and concluded. I suggest the authors to make the abstract more concise and more focusing. I suggest the authors follow a conventional flow line of an abstract with background, issues to be targeted, methods used, results and key conclusions.

**Reply:** Thank you and follow your suggestion. The abstract not repeated here has been revised as the general response.

- Introduction: the introduction looks in a little disorder, so the motivation and novelty of this manuscript looks unclear. It would be better if this part focuses on reviews of typical processes/patterns of under-ice stratification, the differences in QTP lakes (I noticed some results of QTP lakes have been published), and the existing models for lake thermodynamics and their uses for under-ice water. In this way maybe the authors can deliver more clearly your novelty to readers.

**Reply:** Thanks for your constructive advice. The introduction has been revised along the three sections you suggested, and the description about novelty has been added. The revised introduction is shown below:

[revised manuscript text omitted]

- L54: an area larger than 1 km$^2$

**Reply:** The following sentence has been modified accordingly in Introduction:

"The TP is covered by more than 1400 lakes with an area larger than 1 km$^2$"

- L64-69: Please rephrase these sentences to make it more readable.

**Reply:** According to the suggestion about introduction modifications and manuscript length from you and another reviewer, this part is no longer necessary. So, the sentences in L64-69 have been deleted in the revised manuscript.

- L80-86: I suggest to skip over the introduction on unfrozen lakes.

**Reply:** Thanks for your suggestion. The description about unfrozen lakes has been deleted in the revised introduction.

- L89-102: I don't understand how you define the difference of these two types of under-ice water temperature. I guess the water temperature stratification/structure is very dynamic under the ice cover and undergoes some typical stages as depicted/defined in Kirilllin et al (2012), Yang et al (2017, 2021), etc. So maybe it would be better if you introduce the general typical stages of lake temperature stratification through the ice-covered period in boreal, arctic, or temperate lakes, and pointed out the uniqueness or difference of that in highland lakes.

  Kirillin, G., Leppäranta, M., Terzhevik, A., et al, (2012). Physics of seasonally ice-covered lakes: a review. Aquatic sciences, 74(4), 659-682.

**Reply:** Thanks for your suggestion. I have read the articles carefully. The typical stratification stage of lake temperature in ice-covered period is introduced, and the uniqueness of plateau lake temperature is pointed out in the revised introduction as shown in page two.

- L160: 10-m wind speed

**Reply:** The following sentence has been modified accordingly in Section 2.2.1:

"providing meteorological forcing data: wind speed at 10 m, air temperature, specific humidity and air pressure at 2 m, downward shortwave (SR) and longwave radiation (LR) at 1.5 m"

- L168-169: The observed site of water temperature should be added to Figure 1. It would be better to provide basic instrumentation information of water temperature

here, like apparatus, accuracy and frequency, field setup, and sensor depths.

**Reply:** Thank you and follow your advice. The water temperature measurement point (WS) has been added in Figure 1 and the basic instrumentation information has been added in Section 2.2.1:

[Figure]

**Figure 1. (a) Location of Ngoring Lake, the pentagram denotes the lake border station (LBS) and water temperature measurement point (WS). (b) The bathymetry of Ngoring Lake. (b) adapted from (Kirillin et al., 2021).**

"The water temperature measurement site (WS, 35.03º N, 97.70º E, Fig. 1) was located in the northern of Ngoring Lake, where the total water depth was about 26.5 m. The multi-layer water temperature observation system consisted of 16 self-recording RBR SOLO water temperature probes with a precision of 0.01 ℃. The sampling distance and time intervals were 1 m and 10 minutes, respectively."

● L174: to verify the simulated results of what? Surface temperature? As you said, temperature in the MYD11C2 is an 8-day averaged value, so it could lead to uncertainty when you compare your modelling results with MYD11C2. Is there any other product on lake surface temperature that can be used to better evaluate your model results?

**Reply:** The observations of water temperature at WS (Figure 1) are used to verify the simulated multilayer water temperatures. Besides, MODIS data is used to verify the simulated lake surface temperature. There are other lake surface temperature products such as ARC-Lake (Merchant et al., 2013) and AVHRR (Zhu et al., 2022). But by far, the most widely used is MODIS data, even though it is the 8-day composite product and has some bias. In addition, it can show the changing trend of lake surface temperature that can be used to replace observations (Wu et al., 2020; Zhang et al., 2014;

Pour et al., 2012; Song et al., 2016; Xie et al., 2022).

- L384-385: the driving data time step is 30 min, why was the model time step set to 15s?

**Reply:** The 15 s timestep is an internal model timestep (i.e. timestep of the finite-difference scheme, FDS) and it is limited from above by the FDS stability requirement. Specifically, the FDS for k-epsilon turbulence scheme imposes such rigid limitation of the timestep due to its high nonlinearity. In LAKE model, when the time step is smaller than the interval of the driving data, the forcing data will be interpolated according to the time step. But when dt is set to 30 min, the simulation breaks. When the value is set to more than 30 seconds, the simulation result deteriorates. Therefore, we just chose a high time precision and the operation does not take too long 15 s based on the simulations.

- Section 2.2.3: How was the ERA-5 Land data used in this manuscript?

**Reply:** ERA5-Land data was used to extract forcing data for Kilpisjärvi Lake.

- L210-211: Eq (1) seems very complicated, could you please present the physical meaning of each term in the right-hand side?

**Reply:** Thanks for your suggestion. The Eq (1) has been modified and the description about the physical meaning of each term in the right-hand side of the equation has been added in the in Section 3.1.1 of the revised manuscript as follows:

"$c_w \rho_w \frac{\partial T_w}{\partial t} = -c_w \rho_w \frac{1}{A} \int_{\Gamma_A} T_w (u_h \cdot n) dl + \frac{1}{Ah^2} \frac{\partial}{\partial \xi} \left( A K_T \frac{\partial T_w}{\partial \xi} \right) - \frac{1}{Ah} \frac{\partial AS}{\partial \xi} +$

$\frac{1}{Ah} \frac{\partial A}{\partial \xi} \left[ S_b(\xi) + F_{iz,b}(\xi) \right] + \frac{dh}{dt} \frac{\xi}{h} \frac{\partial T_w}{\partial \xi} , \qquad (1)$"

"In Eq. (1), the terms in right-hand respectively represents 1) the advection by inlets, outlets and groundwater discharge, 2) the turbulent diffusion, 3) the divergence of non-turbulent flux, 4) the contribution of the total vertical flux at the sloping bottom, 5) the water budget at the water-air interface."

- Section 3.2: "Validation methods" is not a proper title of this section. This part is actually the method used to evaluate the model accuracy.

**Reply:** Thanks for your advice. The title has been modified to "Methods to evaluate the model accuracy" in Section 3.2.

- (10): the variable symbols should be consistent to Eq. (1), such as ρw , cw . and i is used to denote ice before and ð☐☐¥ð☐☐☐ð☐☐☐ is not mentioned in eq (10).

**Reply:** Thanks for your suggestion. The variable symbols have been consistent to eq. 1. The eq. 8 (original eq. 10) has been modified as follows in Section 3.3:

"The heat storage evolution in water is calculated by the following formulation:
$$Q = c_w \rho_w \sum_{k=1}^{n} T_k \Delta z_k \quad , \qquad (8)$$
where $c_w$ = 4192 J kg$^{-1}$ K$^{-1}$ and $\rho_w$ = 10$^3$ kg m$^{-3}$, $n$ is the layer number, $\Delta z_k$ is depth interval between two successive layers and $T_k$ (K) is the average temperature in layer $k$ (Nordbo et al., 2011; Gan and Liu, 2020)."

- Section 4: Characteristics Analysis is not an informative title, please be specific.

**Reply:** Thanks for your advice. The title has been modified to "Characteristic analysis of water temperature and local climate in two lakes" in Section 4.

- L292: how much is the lowest temperature?

**Reply:** The lowest temperature is 0.47 °C at 2 m. The following has been revised in Section 4.1.1:

"It is pointed out that the under-ice water temperature from 2015 to 2016 in Ngoring Lake rose continuously during the entire ice-covered period according to observations (Wang et al., 2021; Kirillin et al., 2021). In November, the lake mixed evenly with slight oscillation (<1 °C between 2 m and 22 m) and water temperature reduced gradually until the lowest point of 0.47 °C at 2 m on December 12, the lake froze up completely (Fig. 2a). Meanwhile, the air temperature at 2 m fell to -7.79 °C. Ngoring Lake is mostly covered only by bare ice in winter due to drought, less precipitation and snow. In the early ice-covered phase (from December 12 to March 7), the whole lake mixed completely because solar radiation penetrated ice and heated the upper water, which was warm ($< T_{md}$), heavy and sinking (Fig. 2b) (Kirillin et al., 2012). In parallel, water

temperature continued to warm until reached $T_{md}$ on March 7 (Fig. 2a).

In the late ice-covered stage (from March 7 to April 18), the lake stratified. On the one hand, owing to solar radiation strengthened, on the other hand, since radiation absorption of water decayed with depth based on the Beer-Lambert law. Water temperature increased at the rate of 0.052 °C d$^{-1}$ in the layers from 2m to 6 m, which was more rapid than the early stage of 0.035 °C d$^{-1}$. On April 18, the ice melted entirely as well water temperature rose to 5.83 °C at 2 m while remaining at $T_{md}$ below 10 m. After that, full mixing took place rapidly because the lake warmed gaining heat from the sun and atmosphere as a result of ice breaking up (Fig. 2b)."

● Figure 2: the tick spacing of the color bar seems too large so the spatially and temporally fine changes in the water profiles (including the formation and deepening of convective and dicothermal layer) cannot be seen clearly as you described in L302-318.

**Reply:** Thanks for your advice. Figure 2 has been revised with a continuous shading and 0.1 °C interval.

[Figure]

**Figure 2. (a) The daily average water temperature observations of Ngoring Lake at the surface (Ts), 2 m, 9 m, and 22 m from November 1, 2015 to June 1, 2016. Ts is MODIS lake surface temperature. The gray reference lines denote 3.98 °C and 0 °C, respectively. The pink shaded area denotes ice-covered period. The water temperature profile (b) observed and (c) simulated in CTL. The ice-covered period is represented between the two red dashed lines.**

- L307&315: How did you determine the freeze-up and breakup date? From visual observation or remote sensing image (MODIS?)?

**Reply:** The dates were determined by lake temperature observations. The ice thickness at the observation point was not more than 1 m. The freeze-up date was December 12 when the 2 m under-ice water temperature decreased to the lowest point 0.47 °C. The value is closing to the freezing point. The breakup date was April 18 when the 2 m lake temperature drops from maximum value 5.83 °C to 3.88 °C because the ice melted completely, lake releasing heat into atmosphere.

- Section 4.1.2: Do we have to explain here why the weather conditions are different or similar in the two lakes using complicated geographic or geo-statistic experiences? This is not the key point. I think it would be the best if you present general results on the lake information, ice processes, and the water stratification dynamics through the whole ice season, and more importantly, their differences with that of Ngoring Lake.

**Reply:** Thanks for your question and suggestion. It is not important that using complicated geographic or geo-statistic experiences to explain here why the weather conditions are different or similar in the two lakes. This part has been deleted in Section 4.1.2.

Due to the lack of lake temperature of Kilpisjärvi Lake, its under-ice water temperature remained stable during winter from 1992 to 1993 based on Tolonen, 1998. However, it was enough to see the biggest difference between the lake and Ngoring Lake. The characteristic of Ngoring Lake has been pointed out in the description about novelty in page two. The characteristic of Kilpisjärvi Lake as follows:

Kilpisjärvi Lake (69.05º N, 20.83º E, 473 m a.s.l.) is an Arctic tundra lake with average

depth of 19.5 m and maximum depth of 57 m. The lake has a surface area of 37 km$^2$. It starts winter nearer 4 °C as a cryostratified lake.

- L409-410: "but the whole ice season was shifted to occur about half a month earlier than observed". Can you say a little more on how the LAKE calculate the freeze-up and breakup date, e.g. in the method section? Can you explain why the model gave half-month earlier freeze-up?

**Reply:** Thanks for your question. The method how the LAKE model calculated the freeze-up and breakup date has been introduced. When the air temperature is less than 0 °C and the surface water temperature drops to the freezing point, the initial ice formatted. When the net radiation of the lake is greater than 0 W m$^{-2}$, hu ice begins to melt, until the ice thickness changed to 0 m when the ice to melt completely.

According to the study of Stepanenko et al. (2019), the results of the κ-ε turbulence parameterization scheme underestimated the heat capacity of the lake in autumn, so the lake surface cooled rapidly compared to the observed value, leading to an earlier freezing period. However, the improvement of model is not within the scope of this study, so there is not much to study how to solve this problem, but I will pay attention to this problem later.

- L453: delete "with CTL"

**Reply:** Thank you and follow your advice. Been modified accordingly in Section 6.2.

- L468: was reflected…, enters…

**Reply:** The following sentence has been modified accordingly in Section 6.2:

"more solar radiation was reflected and absorbed by snow due to more snowfall accumulation."

- L502: please specify here what the opposite effect is, leading to a decreasing water temperature? In figs. 6b,c,e,f,h, all modeled temperature the upper water layer kept

increasing during the ice-covered period.

**Reply:** The opposite effect means less downward LR, lower air temperature and larger wind speed didn't change the warming trend, but affected the amplitude and rate of warming. The word "opposite" is not appropriate. We have deleted the "opposite" and specified the effect as follows in Section 6.2:

"In conclusion, the stronger downward SR and lower precipitation in TP played positive roles in the water temperature warming during the ice-covered period in Ngoring Lake. Less downward LR, lower air temperature, and larger wind speed didn't change the warming trend but affected the warming amplitude and rate. Specific humidity had no significant influence."

- L514-515: delete "When the lake is….Polashenski 2012)" since you didn't consider the snow layer.

**Reply:** Thank you and follow your advice. Been deleted the sentence accordingly in Section 6.3.

- L536-537: I do not understand this sentence. An increment of 0.1 in albedo means an extend of 15-30 d in ice season?

**Reply:** Sorry, this sentence is not explicit due to my original expression that "When the albedo increased from 0.1 to 0.8, the increase was equivalent to 0.1-step, and the ice-covered period was extended for 15-30 days". This sentence does means when the albedo increased from 0.1 to 0.8, the ice-covered period was extended for 15-30 days for every 0.1 increase.

- L558-563: From conventional experiences, with a constant transmitted solar radiation flux, change in light extinction coefficient of water will of course cause changes in water temperature profile, so also you said "The higher was the extinction coefficient of water, the more heat was absorbed by shallow water and the less heat reached deep layer", the top water layer should get a higher temperature maximum because the solar energy is used to heat a thinner water layer.

If you look at the simulated temperature contours of the whole water column, you may find this regime. But from the point of heat balance, when changing the extinction coefficient of water, the increment in water heat storage doesn't change. I think the authors should elaborate this part.

**Reply:** Thanks for your suggestion. From the point of heat balance, when only the extinction coefficient of water changed, the heat of solar radiation entering the water through the ice sheet is unchanged so that the heat storage of lake did not change. It's just that the heat distribution in the vertical direction is changed. The higher was the extinction coefficient of water, the more heat was absorbed by shallow water and the less heat reached deep layer. This part has been elaborated in Section 6.3 as follows:

"In the sensitivity experiment SIM_$E_w$, the water extinction coefficient had just little influence on winter water temperature, which was shown as 3 m water temperature decreases with the increase of $E_w$ (Fig. 8c). The main reason was that when only the extinction coefficient of water changed, the heat of solar radiation entering the water through the ice sheet is unchanged so that the heat storage of lake did not change. It's just that the heat distribution in the vertical direction is changed. The higher was the extinction coefficient of water, the more heat was absorbed by shallow water and the less heat reached deep layer. The phenomenon that the 3 m water temperature decreases with the extinction coefficient increases becomes more and more obvious in the later stage of ice melting."

- Fig 8b: Why were there sharp drops in all scenarios on ~ April 14? In the text, you stated that the ice broke-up on Mar 31-April 1. I guess Fig. 8b showed wrong dates along x-axis.

**Reply:** Thanks for your question. The drawing program has been checked and it is correct. The date March 31 is the simulated melting date when the heat storage decreased (Figure 7, original Figure 8). In the simulated results, the lake was already ice-free on April 14 when the heat storage dropped due to the arrival of cold air (Supplement: Figure 3).

[Figure]

**Figure 7. (a) The initial water temperature profile in the model is set on March 25, 2016 and the corresponding daily average (b) lake heat storage per unit area is simulated. SIM_E1, SIM_E2 and SIM_E3 are three different initial water temperature profiles.**

[Figure]

**Supplement: Figure 3. The daily averaged 2 m air temperature (red), lake surface temperature in CTL (blue), difference between air temperature and lake surface temperature (green).**

- Section 6.4: (1) How did you estimate the turbulent sensible and latent heat fluxes? Based on Monin-Obulhov theory? (2) Since you stated that the water structure or continuous warming of upper water before the ice breakup has lasting influences on turbulent heat fluxes on the following 1-2 months, could you please say more

on why, through what processes? Right after the breakup, based on heat budget and balance of the lake water, can you estimate quantitatively the contribution of heat storage before the breakup to the turbulent heat change? e.g. comparison between the heat storage and the accumulated heat release to the atmosphere by turbulent sensible and latent exchange.

**Reply:** Yes, the turbulent sensible and latent heat fluxes is calculated based on Monin-Obukhov theory in LAKE model.

Except for the vertical profile of water temperature under ice, other initial conditions in SIM_E1, SIM_E2 and SIM_E3 are set the same as CTL. It can be seen that the initial contour line of the lake at a different temperature, time of ice melt completely, then heat reserves in glacial lakes also change, ice melt completely on the same date, the release of heat flux will also change, this change will continue for some time until the lake and the atmosphere reach the same heat balance.

The amount of heat released as a percentage of the heat stored in the lake before ice completely melts has been calculated. The following sentence has been added in Section 6.4:

"The heat released was in the form of sensible heat and latent heat, accounting for 0.060% (CTL), 0.019% (SIM_E1), 0.036% (SIM_E2) and 0.035% (SIM_E3) of the ice-covered heat storage, respectively."

- L646: "where the air temperature is comparable". I guess not all low-altitude northern lakes have comparable air temperature with QTP lakes. Perhaps some lakes have comparable winter-averaged air temperature, the temporal patterns of air temperatures are different (e.g. even in Fig. 4f).

**Reply:** Agree with your opinion that not all low-altitude northern lakes have comparable air temperature with QTP lakes. Due to your suggestion that it is not important to explain why the weather conditions are different or similar in the two lakes using complicated geographic or geo-statistic experiences. And this part which corresponds "comparable" has been deleted. So, the sentence has been changed as follows:

"Our analysis demonstrates a significant increase in lake temperature during the icecovered period in Ngoring, the largest freshwater lake on the TP, with water temperatures exceeding the freshwater maximum density value $T_{md}$. The heating is governed by strong solar radiation, the factor differing alpine lakes on the TP from the low-altitude northern lakes with similar winter air temperature patterns."

- L665-666: again, what is the "negative feedback"? Besides, feedback is not an accurate word here, maybe influence, impact, or contribution is better.

**Reply:** Thank you and follow your suggestion. The feedback is not appropriate, and it has been changed to influence in Section 7.

"Sensitivity simulations with perturbed local climate data confirmed the decisive role of subsurface solar radiation in the water temperature rise and demonstrated strong negative feedback with winter precipitation amount. The downward LR, air temperature, and wind speed had only a minor influence on the water temperature."

- L674-679: this sentence is too long, please rephrase it to be more readable. By the way, what do you mean by "the difference… lasted for 59-67 days…"? do you mean the under-ice temperature profiles have subsequent impacts on turbulent heat exchange at the air-lake interface after the ice breakup?

**Reply:** Thanks for your advice. The differences mean the sensible and latent heat is different under three stable lake temperature profiles and Ngoring Lake after ice breaking. The difference lasted for 59-97 days. The sentence has been modified as follows in Section 7 to make it readable:

"An important consequence of the under-ice solar heat accumulation consisted in increased sensible and latent heat releases in the subsequent open-water phase. According to the model results, the effects on the surface fluxes of Ngoring Lake lasted for 59-97 days after the ice melt and increased the upward latent and sensible surface heat fluxes up to ~80 W m$^{-2}$ and ~50 W m$^{-2}$, respectively."

---

## Author Response (AR2)

Note: the **comments** and **authors' replies** are in font color of **black** and **blue**, respectively. All changes in revised manuscript are highlighted using yellow background.

**RC1**

1、 On line 66 should emphasize low latitude of Tibetan plateau - these lakes get much stronger sunlight than arctic or sub arctic lakes - which I think is key, not just high altitude. Ie a high altitude lake at 55 North won't behave like this, so contrasting upland/apine versus lowland lakes isn't helpful.

**Reply:** Thanks for your constructive suggestion. Both low latitude and high altitude are important for the strong radiation. The sentence has been modified as follows:

"In particular, the largest alpine lake system of the Qinghai-Tibet Plateau (TP), is not only the highest plateau on Earth with an average altitude of 4000-5000 m, but also located in the relatively low-latitude of 26-39º N ensures a high amount of solar radiation."

2、 This should be emphasized when contrasting QT lake at 34.91 N vs K lake at 69.05 N. Major difference isn't elevation - it is the magnitude and timing of winter darkness and solar radiation. I agree lakes are similar size, but should emphasize early that they have different light climate.

**Reply:** Thanks for your detailed consideration. I agree with you that the latitudes, winter darkness and solar radiation should be emphasized early. The below sentences highlighted in yellow background have been added in the Section 2.2.3 and conclusions:

"ERA5-Land data is applied for a comparative analysis of warming mechanisms and thermal conditions in Tibetan ice-covered lakes against those in the Arctic. The reanalysis forcing data for the geographical position 69.05º N, 20.83º E was adopted as "typical" arctic weather conditions, that is, the SR is 0 W m$^{-2}$ during the winter polar night period. Northern Fennoscandia is covered by several lakes characterized by the longest ice-covered period in Western Europe. The largest of these lakes, Kilpisjärvi, has a similar morphometrical feature to Ngoring (average depth 19.5 m, maximum depth 57 m, surface area 37 km$^2$). However, they receive different SR because of different latitudes. The lake has been intensively studied in the last decades (Kirillin et al., 2015, 2018; Leppäranta et al., 2017, 2019). Its under-ice water temperature remained stable during winter from 1992 to 1993 (Tolonen, 1998). In the following,

model experiments forced by the ERA5 weather data (1992-1993) for the Arctic refer to "Kilpisjärvi" runs."

"The heating is governed by strong solar radiation, the factor differing alpine lakes on the high-altitude and relatively low-latitude TP from the high-latitude and low-altitude northern lakes with similar winter air temperature patterns."

3、 On line 233 are units of 66.7 C right?

**Reply:** I am so sorry, that was my incaution. There should not be any units. I have deleted the unit °C.

4、 On line 291 -294 it is briefly mentioned that there is a massive difference (factor of 4) difference in light climate (SR fluxes). This needs to be discussed earlier. How much of this is due to flux being 0 is Arctic night vs higher values at low latitudes in spring.

**Reply:** Thank you and follow your suggestion. The SR difference has been mentioned in introduction and Section 2.2.3 as shown in comment 2 modification.

5、 Line 408 has conclusion that strong SR fluxes is major difference between sites, which should be foreshadowed earlier in manuscript, (as I mention above).

**Reply:** Thank you and follow your suggestion. They have been foreshadowed in the introduction:

"While the major prerequisite for the ice cover development is sufficient long season with air temperature below the freezing point of water, the heat budget of ice-covered lakes varies with latitude and altitude, depending strongly on the available solar radiation, the latter being the major source of heat for under-ice lake water (Kirillin et al., 2012).During the polar night in the Arctic and temperate lakes covered by snow, the solar heating is minor and the bottom sediment is the main heat source (Winter I according to Kirillin et al., 2012); at later stages of the ice season (Winter II), as the snow melts, solar radiation becomes to the main heat source governing thermal stratification and mixing under ice and the melting process at the ice base (Kirillin et al., 2018, 2020)."

6、 If you deleted the discussion of Arctic lake, I am not sure you'd change conclusion, but it might simplify discussion.

**Reply:** Thanks for your advice and I really took the suggestion seriously. However, the discussion of Arctic lake was not revised. That is because there are only two parts discussed about Arctic lake. The first part is Section 2.2.3 which introduced the Arctic lake and why chose the lake. The other part is Section 4.2 which compared the climate differences and concluded the SR is strong in the high-altitude and relatively low-latitude Tibet. Perhaps they should be retained in order to make the differences between low- and high- latitudes clearer.

7、 DO you say why the Tibetan lake starts off winter with homogenous profile? I assume it is like Nam Co, and is a wide windy lake that is cryomictic. This phrase really just means how the winter stratification starts, not how it evolves, so I'd change phrasing on line 87 - 88 that "nor can be characterized in terms of cryomictic/cryostratified conditions." I think it is cryomictic, and new feature you show is what I thought Kirillin had described as a Winter III period , when high SR fluxes lift under ice temperatures above 4C.

**Reply:** I mentioned Ngoring Lake started off winter with homogenous profile, but did not explain why. The sentence has been revised in Section 4.1:

"In November, because of the strong wind and wide surface, the lake mixed evenly with slight oscillation (<1 °C between 2 m and 22 m) and water temperature decreased gradually until the lowest point of 0.47 °C at 2 m on December 12, the lake froze up completely (Fig. 2a)."

I have carefully read and thought the definition of cryomictic in Yang's article again. I agree with you that it means how the lake stratified at ice forming not how evolved and Ngoring Lake is a cryomictic lake because its wide surface and strong wind speed. Moreover, as you said, Ngoring Lake belongs to the situation that Winter II dominates all ice-covered period, but due to the strong solar radiation, the under-ice water temperature exceeds 4 °C. So, the phrases on line 87-88 have been revised as follows:

"This radiation-dominated regime, is slightly different from the typical heat budget known from earlier studies on ice-covered lakes. Although it belongs to the classification that winter II occupies the whole ice-covered period, the under-ice lake temperature exceeds 4 °C in the late stage due to the strong SR on the TP. Quantification of the resulting heat balance and thermal stratification characteristic of alpine conditions is the subject of the present study."

**Editor**

- Besides adjustments requested by the Editor or Referees, please check your manuscript carefully for typos, missing co-authors and their affiliations, terminology, updates of data in tables, or updates of variables in equations. All these have to be clarified with the Editor and therefore have to be included before you submit your revised manuscript. Should your manuscript be finally accepted it will not be possible to include such rather substantial changes anymore when your manuscript is in final production (proofreading).

**Reply:** Thanks for your suggestions and reviewing very much. I have revised the manuscript based on comments from referees, and checked the manuscript as described above.